# How does overparametrization affect performance on minority groups?

**Saptarshi Roy**                                                 *saptarshi.roy@austin.utexas.edu*
*Department of Computer Science*
*University of Texas at Austin*

**Subha Maity**                                                          *smaity@waterloo.edu*
*Department of Statistics & Actuarial Science*
*University of Waterloo*

**Songkai Xue**                                                             *sxue@umich.edu*
*Department of Statistics*
*University of Michigan, Ann Arbor*

**Mikhail Yurochkin**                                              *mikhail.yurochkin@ibm.com*
*MIT-IBM Watson AI Lab*

**Yuekai Sun**                                                              *yuekai@umich.edu*
*Department of Statistics*
*University of Michigan, Ann Arbor*

**Reviewed on OpenReview:** *https://openreview.net/forum?id=POunezXgvF&referrer*

## Abstract

The benefits of overparameterization for the overall performance of modern machine learning (ML) models are well known. However, the effect of overparameterization at a more granular level of data subgroups is less understood. Recent empirical studies demonstrate encouraging results: (i) when groups are not known, overparameterized models trained with empirical risk minimization (ERM) perform better on minority groups; (ii) when groups are known, ERM on data subsampled to equalize group sizes yields state-of-the-art worst-group accuracy in the overparameterized regime. In this paper, we complement these empirical studies with a theoretical investigation of the risk of overparameterized random feature regression models on minority groups with identical feature distribution as the majority group. In a setting in which the regression functions for the majority and minority groups are different, we show that overparameterization either improves or does not harm the asymptotic minority group performance under the ERM setting when the features are distributed uniformly over the sphere. [1]

## 1 Introduction

Traditionally, the goal of machine learning (ML) is to optimize the *average* or *overall* performance of ML models. The relentless pursuit of this goal eventually led to the development of deep neural networks, which achieve state-of-the-art performance in many application areas. A prominent trend in the development of such modern ML models is overparameterization: the models are so complex that they are capable of perfectly interpolating the data. There is a large body of work showing overparameterization improves the performance of ML models in a variety of settings, *e.g.* ridgeless least squares Hastie et al. (2019), random

---

[1]Codes are available at `https://github.com/smaityumich/overparameterized-group-fairness`.

feature models Belkin et al. (2019); Mei & Montanari (2019) and deep neural networks Nakkiran et al. (2019).

However, as ML models find their way into high-stakes decision-making processes, other aspects of their performance (besides average performance) have been under scrutiny. One aspect that is particularly relevant to the fairness and safety of ML models is their performance on traditionally disadvantaged demographic groups. There is a troubling line of work showing ML models that perform well on average may perform poorly on minority groups of training examples. For example, Buolamwini & Gebru (2018) shows that commercial gender classification systems, despite achieving low classification error on average, tend to misclassify dark-skinned people. In the same spirit, Wilson et al. (2019) shows that pedestrian detection models, despite performing admirably on average, have trouble recognizing dark-skinned pedestrians.

In lieu of this issue, prior literature examines the effect of model size on the generalization error of the worst group. For example, in the simple *classification setting* with spurious correlation, Sagawa et al. (2020) shows that increasing model size beyond the threshold of zero training error (overparameterized regime) could harm the test error for minority groups. In a nutshell, the authors argue that overparameterized models do not effectively learn a structure that generalizes well for both majority and minority groups; instead, they tend to "memorize" the minority points owing to an inductive bias. This is rather surprising as a more extensive set of experiments on benchmark data in Le Pham et al. (2021) reveals that increasing model size either improves or does not harm the worst-group test performance across all settings. In an experiment with California housing prices dataset [2] (see Appendix D), where the target is to predict the median house prices, we find that overparameterization in a random feature regression model (Mei & Montanari, 2019) improves mean squared error for the minority group comprising inland houses. This is in sharp contrast to the findings in Sagawa et al. (2020). Therefore, a comprehensive theoretical investigation is necessary in this area to gain a deeper understanding of the impact of overparameterization on the minority group.

In parallel, Sagawa et al. (2020); An et al. (2021) also shows that subsampling the majority groups is far more successful than upweighting the minority groups in reducing worst-group error. Idrissi et al. (2021) recommends using simple methods, *i.e.* subsampling and reweighting for balanced classes or balanced groups, before venturing into more complicated procedures. They also suggest that newly developed robust optimization approaches for worst-group error control (Sagawa et al., 2019; Liu et al., 2021) could be computationally demanding, and that there is no strong (statistically significant) evidence of advantage over those simple methods.

**Main contributions.** In light of these existing works, we investigate how overparameterization affects the performance of ML models on minority groups in a *regression setup* when the features are generated uniformly over the sphere. In particular, we study the effect of overparameterization under both the usual linear regression and the two-layer random feature (RF) model (Belkin et al., 2019; Mei & Montanari, 2019). As we shall see, *overparamaterization generally improves or stabilizes the generalization error of the minority groups* under the random feature model. On the contrary, the generalization error under the linear regression setup deteriorates with increasing overparameterization. To elaborate more on this, we lay out a more detailed discussion of our contributions below:

1. **Complete picture of limiting ERM risk under RF model:** We develop a simple two-group model for studying the effects of overparameterization on (sub)groups. Our model has parameters controlling signal strength, majority group fraction, overparameterization ratio, discrepancy between the two groups, and error term variance that display a rich set of possible effects. In a high-dimensional asymptotic setting, we develop a fairly complete picture for the limiting risk of the *minimum norm interpolator* under the ERM setup in the two-layer RF regression model.

   In particular, we provide a clear picture of the bias-variance trade-off under this setup as the number of samples $n$ and the number of hidden features $N$ diverge to infinity. For a better understanding of the main result, we present the following informal version of our main theorem:

   **Theorem 1.1** (Informal version of Theorem 3.9)**.** *Let $R_{\mathrm{minor}}$ denote the generalization risk of the minimum norm interpolator (defined in* (2.6)*) on the minority group under the RF model. Under*

[2]https://www.kaggle.com/datasets/camnugent/california-housing-prices

*suitable distributional assumptions on the input and hidden features, and regularity assumptions on the activation function, we have*

$$\lim_{N,n\to\infty} \mathbb{E}[R_{\mathrm{minor}}] = \mathscr{B}^*_{\mathrm{RF}} + \tau^2 \mathscr{V}^*,$$

*where the $\mathscr{B}^*_{\mathrm{RF}}$ is the inductive bias (see Section 3) of the RF model and $\tau^2 \mathscr{V}^*$ is the variance term due to the noise with variance $\tau^2$.*

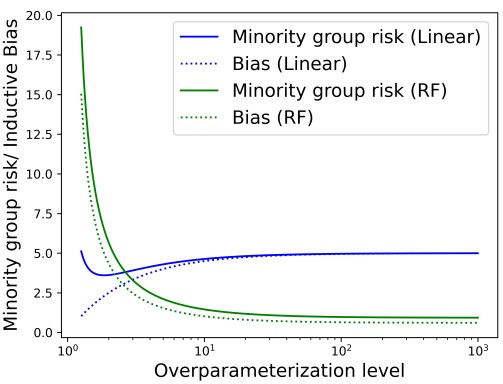 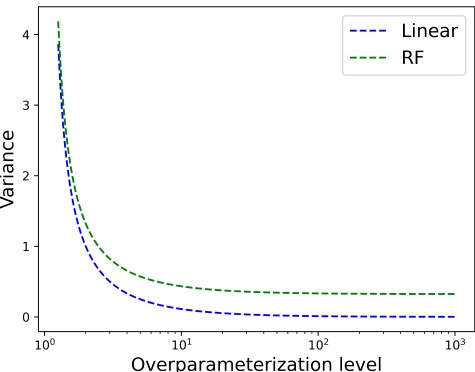

Figure 1: Comparison of Minority group risk, inductive bias, and variance for linear and RF model under 20% minority population. The minority group risk monotonically decreases for the RF model as the overparameterization level increases. However, for the linear model, the trend is increasing under the high overparameterization regime. The trend in variance is similar for both linear and RF models.

Section 3 provides a more detailed explanation of all the quantities. As Figure 1 shows, a more detailed analysis reveals that the bias term $\mathscr{B}^*_{\mathrm{RF}}$ does not increase with the level of overparameterization ($N > n$), which is in sharp contrast to with the linear model case. Also, the trend in the variance is decreasing for both linear and RF models. This shows that the trend of the risk in both cases is essentially influenced by the trend in the inductive bias. As a result, the empirical risk of the RF model in the minority group stabilizes. This phenomenon can be attributed to the strong generalization property of the random feature model due to its highly non-linear architecture. It follows that under a suitable choice of the activation function, the RF model is able to capture or explain all possible directions of the signal that in turn does not allow the bias term to deteriorate with increasing level of overparameterization.

2. **Complete picture of limiting ERM risk under linear model:** We also study the effect of overparameterization in the linear regression model. In contrast to the RF model, we show that the test error for the minority group worsens with the increasing level of overparameterization. In particular, in the case of high signal-to-noise ratio, the prediction accuracy for the minority group worsens with the overparameterization of the model (See Figure 1). In particular, we show that the linear model fails to explain part of the signal that is orthogonal to the input features. This results in a higher inductive bias and it aggravates with the increasing level of overparameterization. This highlights the limitation in generalization power of linear model compared to other non-linear models. We defer this discussion along with theoretical analysis of overparameterized linear models to Appendix A.

3. **Performance of majority group sub-sampling:** In Section 3.1 we also analyze the effect of majority group subsampling on the minority group for the overparameterized RF model. We lay out a concrete theoretical foundation that shows that majority group subsampling improves the minority group performance of the RF model in the overparameterized regime.

Beyond the study of overparameterization in regression tasks with random feature models, we also provide a simulation study for the effect of overparameterization on for classifications with random features. While the details of the simulation settings and the observations are deferred to Appendix E, as a short description of the observations, we see that both the conclusions of Sagawa et al. (2020) and Pham et al. (2021) can be replicated, depending on the simulation setting. If the difference between majority and minority subgroups are small, then model overparameterization generally helps with the performance on minority group, which is the same conclusion found in Pham et al. (2021). On the contrary, for larger differences, model overparameterization harm the performance of minority group, a conclusion that is similar to Sagawa et al. (2020). While a detailed theoretical analysis on classification tasks may reveal the intricate reasons for such distinctions, such analysis is beyond the scope of the paper.

## 1.1 Comparison with prior work

**Overparameterized ML models.** The benefits of overparameterization for *average* or *overall* performance of ML models are well-studied. This phenomenon is known as "double descent", and it asserts that the (overall) risk of ML models decreases as model complexity increases past a certain point. This behavior has been studied empirically Advani & Saxe (2017); Belkin et al. (2018); Nakkiran et al. (2019); Yang et al. (2020) and theoretically Hastie et al. (2019); Montanari et al. (2020); Mei & Montanari (2019); Deng et al. (2020); Liu et al. (2024); Matt & Stöger (2025). All these works focus on the average performance of ML models, while we focus on the performance of ML models on minority groups. The most closely related paper in this line of work is Pham et al. (2021), where they empirically study the effect of model overparameterization on minority risk and find that model overparameterization generally helps or does not harm minority group performance.

**Improving performance on minority groups.** There is a long line of work on improving the performance of models on minority groups. There are methods based on reweighing/subsampling Shimodaira (2000); Cui et al. (2019) and (group) distributionally robust optimization (DRO) Hashimoto et al. (2018); Duchi et al. (2020); Sagawa et al. (2019). The DRO approach has also been adopted in Hu & Chen (2022) to improve fairness in survival analysis. Soma et al. (2025) proposed generalized DRO approach with faster convergence rates and tighter bound over the worst-group errors. In parallel, Peet-Pare et al. (2022) adapt *performative prediction* (Perdomo et al., 2020) to DRO approach for more dynamic environments. Some recent works have also adopted regularization frameworks (An et al., 2022; Zhao et al., 2022), contrastive learning (Park et al., 2022; Zhang et al., 2022), and active learning (Wang et al., 2023) frameworks which uses proxy-sensitive information to induce fairness. In the overparameterized regime, methods based on reweighing are not very effective (Byrd & Lipton, 2019), while subsampling has been empirically shown to improve performance on the minority groups (Sagawa et al., 2020; Idrissi et al., 2021). Our theoretical results confirm the efficacy of subsampling for improving the performance for minority groups and demonstrate that it benefits from overparameterization.

**Group fairness.** literature proposes many definitions of fairness (Chouldechova & Roth, 2018), some of which require similar accuracy on the minority and majority groups (Hardt et al., 2016). Methods for achieving group fairness typically perform ERM subject to some fairness constraints (Agarwal et al., 2018; Kenfack et al., 2023). The interplay between these constraints and overparameterization is beyond the scope of this work.

**Paper organization.** The rest of the paper is organized as follows: In Section 2 we introduce the data-generating model and the random feature model. We also discuss the majority group subsampling scheme under this setup. In Section 3 we present the main theoretical results of the random feature model related to the ERM risk and majority group subsampling along with simulation experiments. Finally, in Section 4 we discuss some of the implications of our work.

## 2 Problem setup

### 2.1 Data generating process

Let $\mathcal{X} \subset \mathbb{R}^d$ be the feature space and $\mathcal{Y} \subset \mathbb{R}$ be the output space. To keep things simple, we consider a two-group setup. Let $P_0$ and $P_1$ be probability distributions on $\mathcal{X} \times \mathcal{Y}$. We consider $P_0$ and $P_1$ as the distribution of samples from the minority and majority groups, respectively. In the minority group, the samples $(x, y) \in \mathcal{X} \times \mathcal{Y}$ are distributed as

$$y \mid x = \beta_0^\top x + \epsilon, \quad \mathbb{E}(\epsilon) = 0, \text{ var}(\epsilon) = \tau^2, \quad x, \epsilon \text{ are independent of each other} \tag{2.1}$$

and $P_X$ is the marginal distribution of the feature $x$, $\beta_0 \in \mathbb{R}^d$ is a vector of regression coefficients, and $\tau^2 \in (0, \infty)$ is the noise level.

In the majority group, the marginal distribution of features is identical, but the conditional distribution of the output is different:

$$y \mid x = \beta_1^\top x + \epsilon, \quad \mathbb{E}(\epsilon) = 0, \text{ var}(\epsilon) = \tau^2, \tag{2.2}$$

where $\beta_1 \in \mathbb{R}^d$ is a vector of regression coefficients *for the majority group*. We note that this difference between the majority and minority groups is a form of *concept drift* or *posterior drift*: the marginal distribution of the feature is identical, but the conditional distribution of the output is different. We focus on this setting because it not only simplifies our derivations, but also isolates the effects of concept drift between subpopulations through the difference $\delta \triangleq \beta_1 - \beta_0$. If the covariate distributions between the two groups are different, then an overparameterized model may be able to distinguish between the two groups, thus effectively modeling the groups separately. In that sense, by assuming that the covariates are equally distributed, we consider the worst-case.

Let $g_i \in \{0, 1\}$ denote the group membership of the $i$-th training sample. The training data $\{(x_i, y_i, g_i)\}_{i=1}^n$ consists of a mixture of samples from the majority and minority groups:

$$\begin{aligned} g_i &\sim \text{Ber}(\pi) \\ (x_i, y_i) \mid g_i &\sim P_{g_i} \end{aligned}, \tag{2.3}$$

where $\pi \in [\frac{1}{2}, 1]$ is the (expected) proportion of samples from the majority group in the training data. We denote $n_1$ as the sample size for majority group in the training data.

### 2.2 Random feature models

Here, we consider a random feature regression model Rahimi & Recht (2007); Montanari et al. (2020)

$$f(x, a, \Theta) = \sum_{j=1}^N a_j \sigma(\theta_j^\top x / \sqrt{d}) \tag{2.4}$$

where $\sigma(\cdot)$ is a non-linear activation function and $\Theta$ is the $N \times d$ weight matrix with $\theta_j$ as the $j$th row. In vector notation, Equation (2.4) can be written as $f(x, a, \Theta) = a^\top z$, where $z = \sigma(\Theta x / \sqrt{d})$. In this vectorized notation, $\sigma(\cdot)$ is applied coordinate wise. The random feature model is similar to a two-layer neural network, but the weights $\theta_j$'s of the hidden layer are set at some (usually random) initial values. In other words, a random feature model fits a linear regression model on the response $y$ using the non-linear random features $\{\sigma(\theta_j^\top x / \sqrt{d})\}_{j \in [N]}$ instead of the original feature vector $x$. These types of models have recently been used Montanari et al. (2020) to provide a theoretical explanation for some of the behaviors observed in large neural network models.

### 2.3 Training of the random feature model

We consider two ways in which a learner may fit the random feature model: empirical risk minimization (ERM) that does not require group annotations; subsampling that does require group annotations. The availability of group annotations is highly application dependant and there is a plethora of prior works considering either of the scenarios (Hashimoto et al., 2018; Zhai et al., 2021; Pham et al., 2021; Sagawa et al., 2019; 2020; Idrissi et al., 2021).

### 2.3.1 Empirical risk minimization (ERM)

The most common way to fit a predictive model is to minimize the empirical risk on the training data:

$$\hat{a} \in \arg\min_{a \in \mathbb{R}^N} \frac{1}{n} \sum_{i=1}^{n} \frac{1}{2} \{y_i - f(x_i, a, \Theta)\}^2$$

$$= \arg\min_{a \in \mathbb{R}^N} \frac{1}{n} \sum_{i=1}^{n} \frac{1}{2} \{y_i - \sum_{j=1}^{N} a_j \sigma(\theta_j^\top x_i/\sqrt{d})\}^2, \tag{2.5}$$

where we recall that the weights $\theta_j$'s of the first layer are randomly assigned. The above optimization (2.5) has a unique solution when the number of random features or neurons is less than the sample size ($N < n$) and we call this regime *underparameterized*. When the number of neurons is greater than the sample size ($N > n$), which we call the *overparameterized* regime, $\hat{a}$ is not unique in (2.5), as there are multiple $a \in \mathbb{R}^N$ that interpolate the training data, *i.e.*, $y_i = \sum_{j=1}^{N} a_j \sigma(\theta_j^\top x_i)$, $i \in [n]$, resulting in a zero training error. In such a situation we set $\hat{a}$ at a specific $a \in \mathbb{R}^N$ which (1) interpolates the training data and (2) has minimum $\ell_2$-norm. This particular solution is known as the *minimum norm interpolant solution* (MNIS) (Hastie et al., 2019; Montanari et al., 2020) and is formally defined as

$$\hat{a}_{\mathsf{mnis}} = \arg\min\{\|a\|_2 : y_i = \sum_{j=1}^{N} a_j \sigma(\theta_j^\top x_i/\sqrt{d}), \ i \in [n]\} = (Z^\top Z)^\dagger Z^\top y, \tag{2.6}$$

where $Z \in \mathbb{R}^{n \times N}$ with $Z_{i,j} = \sigma(\theta_j^\top x_i/\sqrt{d})$ and $A^\dagger$ is the Moore-Penrose inverse of $A$. The minimum norm interpolant solution (2.6) has an alternative interpretation: it is the limiting solution to the ridge regression problem at vanishing regularization strength (Hastie et al., 2019, Section 2.2),

$$\hat{a}_{\mathsf{mnis}} = \lim_{\lambda \to 0+} \hat{a}(\lambda), \ \hat{a}(\lambda) = \arg\min_{a \in \mathbb{R}^N} \left[\frac{1}{n} \sum_{i=1}^{n} \left\{y_i - \sum_{j=1}^{N} a_j \sigma(\theta_j^\top x_i/\sqrt{d})\right\}^2 + \frac{N\lambda}{d} \|a\|_2^2\right]. \tag{2.7}$$

In fact, the conclusion in (2.7) continues to hold in the underparameterized ($N < n$) regime. Hence, we combine the two regimes ($N < n$ and $N > n$) and obtain the ERM solution as (2.7). In the literature Hastie et al. (2019) it is known as the *ridgeless* solution. Finally, having an estimator $\hat{a}$, the response of an individual with feature vector $x$ is predicted to be:

$$\hat{y}(x) = f(x, \hat{a}, \Theta) = \sum_{j=1}^{N} \hat{a}_j \sigma(\theta_j^\top x/\sqrt{d}). \tag{2.8}$$

### 2.3.2 Majority group subsampling

It is known that a model trained by ERM often exhibits poor performance on minority groups Sagawa et al. (2019). One promising way to circumvent the issue is *majority group subsampling*: a randomly drawn subset of training sample points is discarded from majority groups to match the sample sizes for the majority and minority groups in the remaining training data. This emulates the effect of reweighted-ERM, where the samples from minority groups are upweighted in the ERM.

In the underparameterized case, the reweighting is preferred over the subsampling due to its superior statistical efficiency. On the contrary, in the overparameterized case, reweighting may not have the intended effect on the performance of minority groups Byrd & Lipton (2019), but subsampling does (Sagawa et al., 2020; Idrissi et al., 2021). Thus, we consider subsampling as a way of improving performance in minority groups in the overparameterized regime. We note that the subsampling requires the knowledge of the groups/sensitive attributes, so its applicability is limited to problems in which the group identities are observed in the training data.

## 3 Main results for the MNIS under random feature regression model

In this section we focus on the minority group performance of the MNIS estimator $\hat{a}_{\mathsf{mnis}}$ defined in Equation (2.6). From here on, we drop the subscript for notational brevity and simply use $\hat{a}$ to denote the MNIS estimator, and we write the mean squared prediction error for the minority samples in the test data as

$$R_0(\hat{a}) = \mathbb{E}_{x \sim P_X}[\{f(x, \hat{a}, \Theta) - \beta_0^\top x\}^2] = \mathbb{E}_{x \sim P_X}[\{\sum_{j=1}^{N} \hat{a}_j \sigma(\theta_j^\top x/\sqrt{d}) - \beta_0^\top x\}^2], \tag{3.1}$$

Table 1: Table of notations

| Notations | Descriptions |
|---|---|
| $n$ | Total sample size |
| $n_0, n_1$ | Sample-sizes in minority and majority groups |
| $N$ | Number of neurons |
| $d$ | Covariate dimension |
| $\psi_1$ | $\lim_{N,d \to \infty} N/d$ |
| $\psi_2$ | $\lim_{n,d \to \infty} n/d$ |
| $\gamma$ | $\lim_{N,n \to \infty} N/n$ (Overparameterization) |
| $\pi$ | Majority group proportion |
| $\beta_0, \beta_1$ | Regression coefficients for minority and majority groups |
| $\delta$ | $\beta_1 - \beta_0$ |
| $\tau^2$ | Noise variance |

where $x \sim P_X$ is independent of the training data. One may alternatively study $\mathbb{E}_{(x,y) \sim P_0}[\{f(x, \hat{a}, \Theta) - y\}^2]$ as the mean square prediction error for the minority group, but for a fixed noise level $\tau$ it has the same behavior as (3.1), as suggested by the following: $\mathbb{E}_{(x,y) \sim P_0}[\{f(x, \hat{a}, \Theta) - y\}^2] = \mathbb{E}_{x \sim P_X}[\{f(x, \hat{a}, \Theta) - \beta_0^\top x\}^2] + \tau^2$.

A finite sample study of $R_0(\hat{a})$ (3.1) is difficult, in this paper we study its limiting behavior in high-dimensional asymptotic setup. Specifically, recalling the notations from Table 1 we consider $N/d \to \psi_1 > 0$, $n/d \to \psi_2 > 0$ and $n_1/n \to \pi \in [1/2, 1]$. The problem is underparameterized if $\gamma = \psi_1/\psi_2 = \lim_{d \to \infty}\{N/n\} < 1$ and overparameterized if $\gamma > 1$.

To study the minority risk, we first decompose it into several parts and study each of them separately. The first decomposition follows:

**Lemma 3.1.** *We define the bias $\mathscr{B}(\beta_0, \delta, \tau) = \mathbb{E}_{x \sim P_X}[\{\mathbb{E}_\epsilon[f(x, \hat{a}, \Theta)] - \beta_0^\top x\}^2]$ and the variance $\mathscr{V}(\beta_0, \delta, \tau) = \mathbb{E}_{x \sim P_X}[\mathrm{Var}_\epsilon\{f(x, \hat{a}, \Theta)\}]$, where $\epsilon = (\epsilon_1, \ldots, \epsilon_n)^\top$ are the errors in the data generating model, i.e. $\epsilon_i = y_i - x_i^\top \beta_{g_i}$ and $\mathrm{Var}_\epsilon$ denotes the variance with respect to $\epsilon$. The minority risk decomposes as*

$$
\begin{aligned}
&\mathbb{E}_{x \sim P_X, \epsilon}[\{f(x, \hat{a}, \Theta) - \beta_0^\top x\}^2] \\
&= \mathscr{B}(\beta_0, \delta, \tau) + \mathscr{V}(\beta_0, \delta, \tau).
\end{aligned}
\tag{3.2}
$$

The proof is realized by decomposing the square in $[f(x, \hat{a}, \Theta) - \beta_0^\top x]^2 = [\{f(x, \hat{a}, \Theta)\} - \mathbb{E}_\epsilon[f(x, \hat{a}, \Theta)]\} + \{\mathbb{E}_\epsilon[f(x, \hat{a}, \Theta)] - \beta_0^\top x\}]^2$. We now provide an interpretation for the variance term.

**The variance term.** Firstly, we note that the term $\mathscr{V}(\beta_0, \delta, \tau)$ captures the average effect of variance of the prediction rule $f(x, \hat{a}, \Theta)$ with respect to the feature distribution $P_X$. In addition, let $z = \sigma(\Theta x/\sqrt{d})$ as the random feature for new covariates $x$ and note that $\mathscr{V}(\beta_0, \delta, \tau)$ does not depend on the values of $\beta_0$ or $\delta$, as shown in the following display:

$$
\mathrm{Var}_\epsilon\{f(x, \hat{a}, \Theta)\} = \mathrm{Var}_\epsilon\{z^\top \hat{a}\} = \mathrm{Var}_\epsilon\{z^\top (Z^\top Z)^\dagger Z^\top y\} = \mathrm{Var}_\epsilon\{z^\top (Z^\top Z)^\dagger Z^\top \epsilon\}.
\tag{3.3}
$$

Hence we drop $\beta_0$ and $\delta$ from the notation and write the variance term as $\mathscr{V}(\tau)$. Since $\epsilon_i$'s have variance $\tau^2$ we realize that the exact dependence of the variance term with respect to $\tau$ is the following: $\mathscr{V}(\tau) = \tau^2 \mathscr{V}(\tau = 1)$. This also implies that the variance term $\mathscr{V}(\tau)$ *teases out the contribution of data noise level $\tau$ in the minority risk.*

**The inductive bias term.** We notice that the term $\mathscr{B}(\beta_0, \delta, \tau)$ in decomposition (3.2) represents the average bias in prediction for the minority group. We refer to this term as the inductive bias of the RF

model induced by the MNIS estimator. Before we elaborate on this, first note that the bias term does not depend on the noise level $\tau$ (both $\mathbb{E}_\epsilon[f(x, \hat{a}, \Theta)]$ and $\beta_0^\top x$ do not depend on $\tau$) and hence, we denote it as $\mathscr{B}(\beta_0, \delta)$ by dropping $\tau$. Next, we separate out the contributions of $\beta_0$ and $\delta$ in the bias term via a decomposition. For the purpose we recall, $Z = (z_{i,j})_{i \in [n], j \in [N]} \in \mathbb{R}^{n \times N}$ is the matrix of random features and define the following: (i) $X = [x_1^\top, \dots x_n^\top]^\top$ is the covariate matrix in training data, (ii) $X_1$ is the covariate matrix consisting only of the samples from the majority group, and (iii) $Z_1$ is the random feature matrix for the majority group. The decomposition for the bias term follows:

$$\begin{aligned}
\mathscr{B}(\beta_0, \delta) = \mathbb{E}_{x \sim P_X}[\{(z^\top (Z^\top Z)^\dagger Z^\top X - x^\top)\beta_0\}^2 + \{z^\top (Z^\top Z)^\dagger Z_1^\top X_1 \delta\}^2 \\
+ 2(z^\top (Z^\top Z)^\dagger Z^\top X - x^\top)\beta_0 \delta^\top X_1^\top Z_1 (Z^\top Z)^\dagger z].
\end{aligned} \tag{3.4}$$

The first terms on the right-hand side of (3.4) essentially capture the inductive bias due to the misspecification in the random features model, *i.e.*, the part of the signal $x^\top \beta_0$ that cannot be explained by the best fitted RF model. On the other hand, the remaining terms capture the part of the bias arising due to the model misspecification in the two-group model. These two bias terms together contribute to the overall inductive bias in the RF model. Next, we make some assumptions on $\beta_0$ and $\delta$ vectors:

**Definition 3.2.** *The quantities $F_\beta, F_\delta > 0$ and $F_{\beta,\delta} \in \mathbb{R}$ are defined as follows: $F_\beta := \|\beta_0\|_2$, $F_\delta := \|\delta\|_2$ and $F_{\beta,\delta} := \langle \beta_0, \delta \rangle$.*

Here, $F_\beta$ and $F_\delta$ are the $\ell_2$ signal strengths of $\beta_0$ and $\delta$. The decomposition in the next lemma separates out the contributions of $\beta_0$ and $\delta$ in $\mathscr{B}(\beta_0, \delta)$.

**Lemma 3.3.** *Define the following:*

$$\begin{aligned}
\mathscr{B}_\beta &= \mathbb{E}_{x \sim P_X}[\|z^\top (Z^\top Z)^\dagger Z^\top X - x^\top\|_2^2 / d] \\
\mathscr{B}_\delta &= \mathbb{E}_{x \sim P_X}[\|z^\top (Z^\top Z)^\dagger Z_1^\top X_1\|_2^2 / d] \\
\mathscr{C}_{\beta,\delta} &= 2\mathbb{E}_{x \sim P_X}[(z^\top (Z^\top Z)^\dagger Z^\top X - x^\top)X_1^\top Z_1 (Z^\top Z)^\dagger z]
\end{aligned} \tag{3.5}$$

*Then we have $\mathscr{B}(\beta_0, \delta) = F_\beta^2 \mathscr{B}_\beta + F_\delta^2 \mathscr{B}_\delta + F_{\beta,\delta} \mathscr{C}_{\beta,\delta}$.*

In Lemma 3.3 we see that $F_\delta^2 \mathscr{B}_\delta + F_{\beta,\delta} \mathscr{C}_{\beta,\delta}$ quantifies the contribution of the model misspecification in the two-group model, and we call this *misspecification error term*. For the other terms in the decomposition we utilize the results of Montanari et al. (2020) who studied the effect of overparameterization on the overall performance (*i.e.*, in a single group model), thus the misspecification error term does not appear in their analysis. Our theoretical contribution in this paper is studying the *exact asymptotics of the misspecification error terms*. Before we present the asymptotic results we introduce some assumptions and definitions. First we make a distributional assumption on the covariates $x_i$'s and the random weights $\theta_j$'s which allow for easier analysis of the minority risk.

**Assumption 3.4.** *We assume that $\{x_i\}_{i=1}^n$ and $\{\theta_j\}_{j=1}^N$ are i.i.d. $\mathrm{Unif}\{\mathbb{S}^{d-1}(\sqrt{d})\}$, i.e., uniformly over the surface of a $d$-dimensional Euclidean ball of radius $\sqrt{d}$ and centered at the origin.*

Note that when $x_i$ follows an isotropic standard Gaussian distribution, for large $d$ one can approximate the distribution by $\mathrm{Unif}\{\mathbb{S}^{d-1}(\sqrt{d})\}$, because $d^{-1/2}\|x\|_2 \approx 1$. Also, in practice, one can scale the features so that they reside on some scaled version of the unit sphere. The main reason to consider this simplifying assumption is that it lends us certain technical advantages while studying asymptotic convergence results.

Next we assume that activation function $\sigma$ has some properties, which are satisfied by any commonly used activation functions, *e.g.* ReLU and sigmoid activations.

**Assumption 3.5.** *The activation function $\sigma : \mathbb{R} \to \mathbb{R}$ is weakly differentiable with weak derivative $\sigma'$ and for some constants $c_0, c_1 > 0$ it holds $|\sigma(u)|, |\sigma'(u)| \le c_0 e^{c_1 |u|}$.*

Both assumptions 3.4 and 3.5 also appear in Montanari et al. (2020). Note that the weak differentiability condition in Assumption 3.5 is fairly general and is enjoyed by standard activation functions such as RELU or soft-max activation function. Below we define some quantities which we require to describe the asymptotic results.

**Definition 3.6.** *We define the following quantities:*

1. *For the constants*

$$\mu_0 = \mathbb{E}\{\sigma(G)\}, \qquad \mu_1 = \mathbb{E}\{G\sigma(G)\}, \qquad \mu_\star^2 = \mathbb{E}\{\sigma(G)^2\} - \mu_0^2 - \mu_1^2, \qquad (3.6)$$

*where the expectation is taken with respect to $G \sim \mathrm{N}(0,1)$ we assume that $0 < \mu_0, \mu_1, \mu_\star < \infty$ and define $\xi = \mu_1/\mu_\star$.*

2. *Recall that $N/d \to \psi_1 > 0$, $n/d \to \psi_2 > 0$ and $n_1/n \to \pi \in [1/2, 1]$. We set $\psi = \min\{\psi_1, \psi_2\}$ and define*

$$\chi = -\frac{[(\psi\xi^2 - \xi^2 - 1)^2 + 4\xi^2\psi]^{1/2} + (\psi\xi^2 - \xi^2 - 1)}{2\xi^2}.$$

3. *We furthermore define the following:*

$$\begin{aligned}
\mathscr{E}_0^\star &= -\chi^5\xi^6 + 3\chi^4\xi^4 + (\psi_1\psi_2 - \psi_1 - \psi_2 + 1)\chi^3\xi^6 - 2\chi^3\xi^4 - 3\chi^3\xi^2 \\
&\quad + (\psi_1 + \psi_2 - 3\psi_1\psi_2 + 1)\chi^2\xi^4 + 2\chi^2\xi^2 + \chi^2 + 3\psi_1\psi_2\chi\xi^2 - \psi_1\psi_2, \\
\mathscr{E}_1^\star &= \psi_2\chi^3\xi^4 - \psi_2\chi^2\xi^2 + \psi_1\psi_2\chi\xi^2 - \psi_1\psi_2, \\
\mathscr{E}_2^\star &= \chi^5\xi^6 - 3\chi^4\xi^4 + (\psi_1 - 1)\chi^3\xi^6 + 2\chi^3\xi^4 + 3\chi^3\xi^2 + (-\psi_1 - 1)\chi^2\xi^4 \\
&\quad - 2\chi^2\xi^2 - \chi^2.
\end{aligned}$$

Equipped with the assumptions and the definitions we are now ready to state the asymptotics for each of the terms in the minority group prediction errors. The following lemma, which states the asymptotic results for $\mathscr{B}_\beta$ and $\mathscr{V}(\tau)$, has been proven in Montanari et al. (2020).

**Lemma 3.7** (Theorem 5.7, Montanari et al. (2020))**.** *Let the assumptions 3.2, 3.4 and 3.5 hold. Define $\mathscr{B}^\star = \mathscr{E}_1^\star/\mathscr{E}_0^\star$ and $\mathscr{V}^\star = \mathscr{E}_2^\star/\mathscr{E}_0^\star$ where $\mathscr{E}_0^\star$, $\mathscr{E}_1^\star$ and $\mathscr{E}_2^\star$ are defined in Definition 3.6. Then the following hold:*

$$\lim_{d\to\infty} \mathbb{E}[\mathscr{B}_\beta] = \mathscr{B}^\star, \quad \lim_{d\to\infty} \mathbb{E}[\mathscr{V}(\tau)] = \tau^2\mathscr{V}^\star,$$

*where the expectation $\mathbb{E}$ is taken over $\{x_i\}_{i=1}^n$, $\{\theta_j\}_{j=1}^N$.*

**Trend in the variance term.** We again recall that the variance term $\mathscr{V}(\tau)$ teases out the contributions of noise $\epsilon$ in the minority group prediction error. Here, the trend that we're most interested in is the effect of overparameterization: how does the asymptotic $\mathscr{V}^\star$ behave as a function of $\gamma = \psi_1/\psi_2 = \lim_{n,N\to\infty}(N/n)$ when $\psi_2 = \lim_{n,d\to\infty}(n/d)$ is held fixed, and $\gamma > 1$. Although it is difficult to understand such trends from the mathematical definitions of $\mathscr{V}^\star$ we notice in Figure 2 (left) that the variance decreases in overparameterized regime ($\gamma > 1$) with increasing model size ($\gamma$).

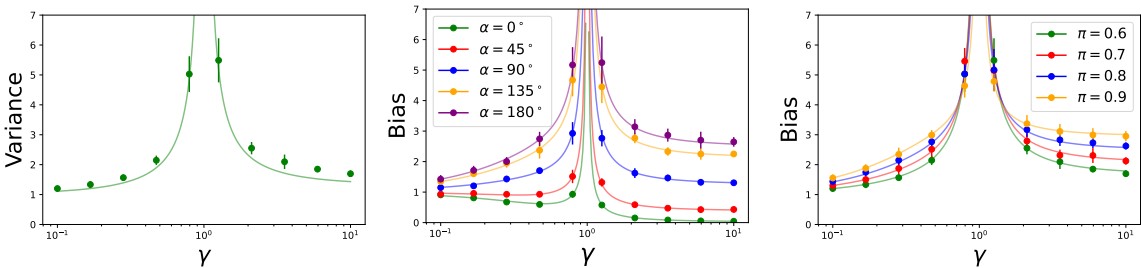

Figure 2: Trends for bias and variance in the minority ERM risk for varying overparameterization $\gamma$ with $d = 200$ and $n = 400$. *Left:* variances for varying $\gamma$. *Middle:* biases for different $\gamma$ and $\alpha$, where $\alpha$ is the angle between $\beta_0$ and $\beta_1$. Here the majority group proportion $\pi$ is set at 0.8. *Right:* biases for different $\pi$'s and $\gamma$'s when $\alpha$ is set at 180°. The solid lines are the theoretical predictions and the points with error-bars represent the empirical predictions over 10 random runs.

Next, we study the asymptotics of $\mathscr{B}_\delta$ and $\mathscr{C}_{\beta,\delta}$ which, as discussed in lemma 3.3, quantify the part of the minority group prediction risk that comes from the model misspecification in the two group model (2.3).

**Lemma 3.8** (Misspecification error terms). *Let the assumptions 3.2, 3.4 and 3.5 hold. Following the definitions of $\mathscr{B}^\star$ and $\mathscr{V}^\star$ in lemma 3.7 we further define $\Psi_2^\star = \mathscr{B}^\star - 1 + 2(\chi + \psi)$, where $\chi$ and $\psi$ are defined in Definition 3.6. Then*

$$\lim_{d \to \infty} \mathbb{E}[\mathscr{B}_\delta] = \mathscr{M}_1^\star \triangleq \pi(1-\pi)\mathscr{V}^\star + \pi^2 \Psi_2^\star, \quad \lim_{d \to \infty} \mathbb{E}[\mathscr{C}_{\beta,\delta}] = \mathscr{M}_2^\star \triangleq \pi(\mathscr{B}^\star - 1 + \Psi_2^\star)$$

*where as in lemma 3.7, the expectation $\mathbb{E}$ is is taken over $\{x_i\}_{i=1}^n$, $\{\theta_j\}_{j=1}^N$.*

Below we describe the observed trends in the bias term in the minority risk.

**Trends in the bias term.** Here, we study the asymptotic trend for the bias term $\mathscr{B}(\beta_0, \delta)$. We see from the bias decomposition in Lemma 3.3 and the term by term asymptotics in Lemmas 3.7 and 3.8 that the bias term asymptotically converges to $F_\beta^2 \mathscr{B}^\star + F_\delta^2 \mathscr{M}_1^\star + F_{\beta,\delta} \mathscr{M}_2^\star$. We mainly study the trend of this bias term in overparameterized regime with respect to the three parameters: (1) the overparametrization parameter $\gamma$, (2) the majority group proportion $\pi$ and (3) the angle between the vectors $\beta_0$ and $\beta_1$, which is denoted by $\alpha$. The middle and right panels in Figure 2 give an overall understanding of the trend of the bias term with respect to the aforementioned parameters. In both of these figures we set $\|\beta_1\|_2 = \|\beta_0\|_2 = 1$. In the middle panel, we see that for a fixed angle $\alpha$, the bias generally decreases with overparameterization. In contrast, for a fixed level of overparameterization $\gamma$, the bias increases as the angle $\alpha$ increases. This is not surprising as intuitively, as the angle $\alpha$ grows, the separation between $\beta_0$ and $\beta_1$ becomes more prominent. As a consequence, the accuracy of the model worsens and that is being reflected in the plot of the bias term.

Similar trend can also be observed when the majority group proportion $\pi$ varies for a fixed angle $\alpha$ (right most panel of Figure 2). Here we set $\beta_1 = -\beta_0$, i.e., the angle $\alpha = 180°$. As expected, the bias increases with parameter $\pi$ for a fixed $\gamma$ because of higher imbalance in the population. Whereas, for a fixed $\pi$, the bias decreases with increasing $\gamma$. Thus, in summary, we can conclude that overparameterization generally improves the worst group performance of the model for fixed instances of $\alpha$ and $\pi$.

In all the plots in Figure 2, we also show the empirical variances and biases along with their error bars computed over 10 repetitions. Within each repetition, a random features model was trained with a dataset generated according to the model described in Section 2.1. The sample sizes and remaining data generating configurations, such as $d$, $\pi$, $\alpha$, and $\gamma$, of the training datasets are described within Figure 2. We then calculated their mean variances using Equation (3.3) and biases using Equation (3.4) over a test dataset with sample size 500 generated according to the same distribution. The same strategy is also used for Figure 3.

The following theorem summarizes the asymptotic results in Lemmas 3.7 and 3.8 and states the asymptotic for minority group prediction error.

**Theorem 3.9.** *Let the assumptions 3.4 and 3.5 hold. Following the definitions of $\mathscr{B}^\star$, $\mathscr{V}^\star$, $\mathscr{M}_1^\star$ and $\mathscr{M}_2^\star$ in Lemmas 3.7 and 3.8, we have the term by term asymptotics:*

$$\lim_{d \to \infty} \mathbb{E}[\mathscr{B}_\beta] = \mathscr{B}^\star, \ \lim_{d \to \infty} \mathbb{E}[\mathscr{V}(\tau)] = \tau^2 \mathscr{V}^\star,$$
$$\lim_{d \to \infty} \mathbb{E}[\mathscr{B}_\delta] = \mathscr{M}_1^\star, \ and \ \lim_{d \to \infty} \mathbb{E}[\mathscr{C}_{\beta,\delta}] = \mathscr{M}_2^\star,$$

*and the minority group prediction error has the following asymptotic*

$$\mathbb{E}[R_0(\hat{a})] = F_\beta^2 \mathbb{E}[\mathscr{B}_\beta] + F_\delta^2 \mathbb{E}[\mathscr{B}_\delta] + F_{\beta,\delta} \mathbb{E}[\mathscr{C}_{\beta,\delta}] + \tau^2 \mathbb{E}[\mathscr{V}(\tau = 1)]$$
$$\to F_\beta^2 \mathscr{B}^\star + F_\delta^2 \mathscr{M}_1^\star + F_{\beta,\delta} \mathscr{M}_2^\star + \tau^2 \mathscr{V}^\star.$$

A complete proof of the theorem is provided in Appendix B.

**Trend in minority group prediction error for ERM.** To understand the trends in minority group prediction error we combine the trends in the bias and the variance terms. We again recall that we're interested in the effect of overparameterization, *i.e.* growing $\gamma$ when $\gamma > 1$. In the discussions after Lemmas 3.7 and 3.8 (and in Figure 2) we notice that both the bias and variance terms decrease or do not increase with

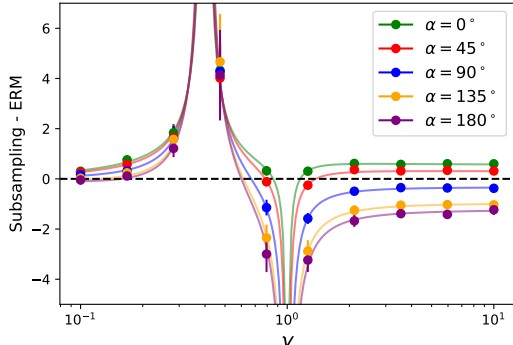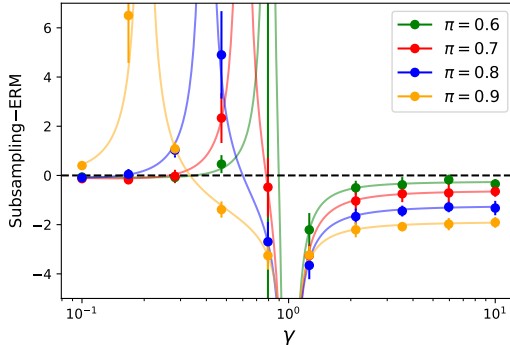

Figure 3: Difference in minority risk between subsampling and ERM with $d = 200, n = 400$. *Left*: the differences in minority risks for several values of $\alpha$ (the angle between $\beta_0$ and $\beta_1$) and overparameterzation $\gamma$, where majority proportion $\pi$ is set at 0.8. *Right*: the minority risk differences for several values of $\pi$ and $\gamma$ when $\alpha$ is set at 180°.

growing overparameterization for any choices of the majority group proportion $\pi$ and the angle $\alpha$ between $\beta_0$ and $\beta_1$. Since the minority risk decomposes to bias+variance terms (3.2), we notice that similar trends hold for the overall minority risk. In other words, *overparameterization improves or does not harm the minority risk*. These trends agree with the empirical findings in Pham et al. (2021).

## 3.1 Limiting risk for majority group subsampling

Though overparameterization of ML models does not harm the minority risk for ERM, they generally produce large minority risk Pham et al. (2021). To improve the risk over minority group Sagawa et al. (2020); Idrissi et al. (2021) recommend using majority group subsampling before fitting ERM and show that they achieve state-of-the-art minority risks. In addition, An et al. (2021) also theoretically showed that resampling produces superior results while training ML models under simple optimization setup. However, the authors do not investigate the asymptotic test error under more complicated model. Here, we complement these prior works with a theoretical study on the minority risk for random feature models in the two group regression setup (2.3). We recall from subsection 2.3.2 that subsampling discards a randomly chosen subsample of size $n_1 - n_0$ from the majority group (of size $n_1$) to match the sample size ($n_0$) of the minority group, and then fits an ERM over the remaining sample points. Here, we repurpose the asymptotic results for ERM (Theorem 3.9) to describe the asymptotic behavior for subsampling by carefully reviewing the changes in the parameters $\psi_1$, $\psi_2$ and $\pi$. First we notice that the total sample size after subsampling is $n_S = 2n_0$ (both the majority and minority groups have sample size $n_{0,S} = n_{1,S} = n_0$) which means the new majority sample proportion is $\pi_S = n_{0,S}/n_S = 1/2$. We further notice that $\psi_1$ remains unchanged, *i.e.*, $\psi_{1,S} = \psi_1$ whereas $\psi_2 = \lim_{d\to\infty} n/d$ changes to $\psi_{2,S} = \lim_{d\to\infty} n_S/d = \lim_{d\to\infty} 2n_0/d = \lim_{d\to\infty} 2(n_0/n) \times (n/d) = 2(1-\pi)\psi_2$. The subsampling setup is underparameterized when $n_S = 2n_0 > N$ or $\gamma_S = \psi_{1,S}/\psi_{2,S} = \gamma/(2*(1-\pi)) < 1$ and overparameterized when $\gamma_S > 1$. Rest of the setup in subsampling remains exactly the same as in the ERM setup, and the results in Lemmas 3.7, 3.8 and Theorem 3.9 continue to hold with the new parameters: $\psi_{1,S}$, $\psi_{2,S}$ and $\pi_S$.

**ERM vs subsampling.** We compare the asymptotic risks of ERM and majority group subsampling in overparameterized setting. Figure 3 shows the trend of the error difference between subsampling and ERM. Similar to the previous discussion, we denote by $\alpha$ the angle between $\beta_1$ and $\beta_0$, and set $\|\beta_1\|_2 = \|\beta_0\|_2 = 1$. In the left panel, we fix the majority group parameter $\pi = 0.8$ and vary $\alpha$. The empirical quantities are also evaluated in the similar fashion as in Figure 2. It is generally observed that subsampling improves the worst group performance over ERM, which is consistent with the empirical findings in Sagawa et al. (2020) and Idrissi et al. (2021). Moreover, the improvement is prominent when the group regression coefficients are well separated, i.e., $\alpha$ is large. Intuitively, when $\alpha$ is very large, the distinction between majority and

minority groups is more noticeable. As a consequence, the effect of under representation on minority group becomes more relevant, due to which the worst group performance of ERM is affected. Whereas, subsampling alleviates this effect of under representation by homogenizing the group sizes. On the other hand, the effect of under representation becomes less severe when $\beta_0$ and $\beta_1$ are close by, i.e., $\alpha$ is small. In this case, we see subsampling does not improve the worst group performance significantly. In fact, for larger values of $\gamma$, subsampling performs slightly worse compared to ERM. This is again not surprising, as for smaller separation, the population structure of the two groups becomes more homogeneous. Thus, full sample ERM should deliver better performance compared to subsampling.

Similar trend is also observed in the right panel where we fix $\alpha = 180°$ and vary $\pi$. Again, improvement due to subsampling is most prominent for larger values of $\pi$, which corresponds to greater imbalance in the data. Unlike the previous case, the subsampling always helps in terms of worst group performance but the improvement becomes less noticeable with decreasing $\pi$ as overparameterization grows.

## 4 Summary and discussion

In this paper, we studied the performance of overparameterized ML models on minority groups. We set up a two-group model and derived the limiting risk of random feature models in the minority group (see Theorem 3.9). In our theoretical finding, we generally see that overparameterization improves or does not harm the minority risk of ERM, which complements the findings in Pham et al. (2021). We also show theoretically that majority group subsampling is an effective way of improving the performance of overparameterized models in minority groups. This confirms the empirical results on subsampling overparameterized neural networks given in Sagawa et al. (2020) and Idrissi et al. (2021).

**Assumption implications.** In our study, an important assumption is that the feature distributions between the two groups are the same (Concept drift, see Section 2.1). This is the worst case for the minority group because an overparameterized model can not leverage any difference in the feature distributions between the groups to improve minority group performance. It is encouraging that overparameterization does not hurt the minority group even under this worst-case setting. Extending our results to the two-group setup with a shift in feature distribution is an interesting future work direction.

**Practical implications** One of the main conclusions of this paper is that overparameterization generally helps or does not harm the worst group performance of ERM. In other words, using overparameterized models is unlikely to magnify disparities across groups. However, we warn the practitioners that overparameterization *should not* be confused with a method for improving the minority group performance. Dedicated methods such as group distributionally robust optimization (Sagawa et al., 2019) and subsampling Sagawa et al. (2020); Idrissi et al. (2021) are far more effective. In particular, subsampling improves worst-group performance and benefits from overparameterization as demonstrated in prior empirical studies Sagawa et al. (2020); Idrissi et al. (2021) and in this paper.

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

# A    Ridgeless regression

Similar to Section 3, here we describe analyzing the linear regression model in the overparameterized regime. The learner fits a linear model

$$f(x) \triangleq \beta^T x$$

to the training data, where $\beta \in \mathbb{R}^d$ is a vector of regression coefficients. We note that although the linear model is well-specified for each group, it is *misspecified* for the mixture of two groups. The learner fits a linear model to the training data in one of two ways: (1) empirical risk minimization (ERM) and (2) subsampling.

## A.1    Model fitting

**Empirical risk minimization (ERM)**    The most common way of fitting a linear model is ordinary least squares (OLS):

$$\widehat{\boldsymbol{\beta}}_{\text{OLS}} \in \arg\min_{\beta \in \mathbb{R}^d} \tfrac{1}{n} \sum_{i=1}^n \tfrac{1}{2}(y_i - \beta^T x_i)^2 = \arg\min_{\beta \in \mathbb{R}^d} \tfrac{1}{2n}\|y - X\beta\|_2^2,$$

where the rows of $X \in \mathbb{R}^{n \times d}$ are the $x_i$'s and the entries of $y \in \mathbb{R}^n$ are the $y_i$'s. We note that ERM does not use the sensitive attribute (even during training), so it is suitable for problems in which the training data does not include the sensitive attribute.

In the overparameterized case ($n < d$), $\widehat{\boldsymbol{\beta}}_{\text{OLS}}$ is not well-defined because there are many linear models that can interpolate the training data. In this case, we consider the minimum $\ell_2$ norm (min-norm) linear model with zero training error:

$$\widehat{\boldsymbol{\beta}}_{\min} \in \arg\min\{\|\beta\|_2 \mid y = X\beta\} = (X^T X)^\dagger X^T y.$$

where $(X^T X)^\dagger$ denotes the Moore-Penrose pseudoinverse of $X^T X$. The min-norm least squares estimator is also known as the ridgeless least squares estimator because $\widehat{\boldsymbol{\beta}}_{\min} = \lim_{\lambda \searrow 0} \widehat{\boldsymbol{\beta}}_\lambda$, where

$$\widehat{\boldsymbol{\beta}}_\lambda \in \arg\min_{\beta \in \mathbb{R}^d} \tfrac{1}{n}\|y - X\beta\|_2^2 + \lambda\|\beta\|_2^2.$$

We note that if $X$ has full column rank (*i.e.* $X^T X$ is non-singular), then $\widehat{\boldsymbol{\beta}}_{\min}$ is equivalent to $\widehat{\boldsymbol{\beta}}_{\text{OLS}}$.

**Majority group subsampling**    It is known that models trained by ERM may perform poorly on minority groups Sagawa et al. (2019). One promising way of improving performance on minority groups is majority group subsampling–randomly discarding training samples from the majority group until the two groups are equally represented in the remaining training data. This achieves an effect similar to that of upweighing the loss on minority groups in the underparameterized regime.

In the underparameterized case ($n > d$), reweighing is more statistically efficient (it does not discard training samples), so subsampling is rarely used. On the other hand, in the overparameterized case, reweighing may not have the intended effect on the performance of overparameterized models in minority groups Byrd & Lipton (2019), but Sagawa et al. (2020) show that subsampling does. Thus we also consider subsampling here as a way of improving performance in minority groups. We note that reweighing requires knowledge of the sensitive attribute, so its applicability is limited to problems in which the sensitive attribute is observed during training.

## A.2    Limiting risk of ERM and subsampling

We are concerned with the performance of the fitted models on the minority group. In this paper, we measure performance on the minority group with the mean squared prediction error on a test sample from the minority group:

$$R_0(\beta) \triangleq \mathbb{E}\big[((\beta - \beta_0)^\top x)^2 \mid X\big] = \mathbb{E}\big[(\beta - \beta_0)^\top \Sigma (\beta - \beta_0) \mid X\big]. \tag{A.1}$$

We note that the definition of (A.1) is conditional on $X$; *i.e.* the expectation is with respect to the error terms in the training data and the features of the test sample. Although it is hard to evaluate $R_0$ exactly for

finite $n$ and $d$, it is possible to approximate it with its limit in a high-dimensional asymptotic setup. In this section, we consider an asymptotic setup in which $n, d \to \infty$ in a way such that $\frac{d}{n} \to \gamma \in (0, \infty)$. If $\gamma < 1$, the problem is underparameterized; if $\gamma > 1$, it is overparameterized.

To keep things simple, we first present our results in the special case of isotropic features. We start by formally stating the assumptions on the distribution of the feature vector $P_X$.

**Assumption A.1.** *The feature vector $x \sim P_X$ has independent entries with zero mean, unit variance, and finite $8 + \epsilon$ moment for some $\epsilon > 0$.*

### A.3 Empirical risk minimization

We start by decomposing the minority risk (A.1) of the OLS estimator $\widehat{\boldsymbol{\beta}}_{\text{OLS}}$ and the ridgeless least squares estimator $\widehat{\boldsymbol{\beta}}_{\min}$. Let $n_0$ and $n_1$ be the number of training examples from the minority and the majority groups respectively. Without loss of generality, we arrange the training samples so that the first $n_0$ examples are those from the minority group:

$$ X = \begin{bmatrix} X_0 \\ X_1 \end{bmatrix}, \quad y = \begin{bmatrix} y_0 \\ y_1 \end{bmatrix}. $$

**Lemma A.2** (ERM minority risk decomposition). *Let $\widehat{\boldsymbol{\beta}}$ be either $\widehat{\boldsymbol{\beta}}_{OLS}$ or $\widehat{\boldsymbol{\beta}}_{\min}$. We have*

$$
\begin{aligned}
R_0(\widehat{\boldsymbol{\beta}}) &= \mathbb{E}\big[(\widehat{\boldsymbol{\beta}} - \beta_0)^\top (\widehat{\boldsymbol{\beta}} - \beta_0)\big] \\
&= \beta_0^\top (I_d - \Pi_X)\beta_0 + \frac{\tau^2}{n} Tr(\widehat{\boldsymbol{\Sigma}}^\dagger) + \delta^\top \Big\{ \begin{bmatrix} 0_{n_0 \times d} \\ X_1 \end{bmatrix}^\top X/n \Big\} (\widehat{\boldsymbol{\Sigma}}^\dagger)^2 \Big\{ X^\top \begin{bmatrix} 0_{n_0 \times d} \\ X_1 \end{bmatrix} /n \Big\} \delta \\
&\quad + 2\delta^\top \Big\{ \begin{bmatrix} 0_{n_0 \times d} \\ X_1 \end{bmatrix}^\top X/n \Big\} (\widehat{\boldsymbol{\Sigma}}^\dagger)^2 \widehat{\boldsymbol{\Sigma}} \beta_0
\end{aligned}
\tag{A.2}
$$

*where $\Pi_X \triangleq X^\dagger X$ is the projector onto $\mathsf{ran}(X^\top)$, $\delta \triangleq \beta_0 - \beta_1$, and $\widehat{\boldsymbol{\Sigma}} \triangleq \frac{1}{n} X^\top X$ is the sample covariance matrix of the features in the training data.*

**Inductive bias** We recognize the first term on the right side of (A.2) as a squared bias term. This term reflects the inductive bias of ridgeless least squares: it is orthogonal to $\mathsf{ker}(X)$, so it cannot capture the part of $\beta_0$ in $\mathsf{ker}(X)$. We note that this term is only non-zero in the overparameterized regime: $\widehat{\boldsymbol{\beta}}_{\text{OLS}}$ has no inductive bias.

**Lemma A.3** (Hastie et al. (2019), Lemma 2). *In addition to Assumption A.1, assume $\|\beta_0\|_2^2 = s_0$ for all $n, d$. We have*

$$ \beta_0^\top (I_d - \Pi_X)\beta_0 \xrightarrow{p} \big\{ s_0 \big(1 - \tfrac{1}{\gamma}\big) \big\} \vee 0 \text{ as } n, d \to \infty, \tfrac{d}{n} \to \gamma. $$

**Variance** The second term on the right side of (A.2) is a variance term. The limit of this term in the high-dimensional asymptotic setting is known.

**Lemma A.4** (Hastie et al. (2019), Theorem 1, Lemma 3). *Under Assumption A.1, we have*

$$ \frac{\tau^2}{n} Tr(\widehat{\boldsymbol{\Sigma}}^\dagger) \xrightarrow{p} \begin{cases} \tau^2 \frac{\gamma}{1-\gamma} & \gamma < 1 \\ \frac{\tau^2}{\gamma-1} & \gamma > 1 \end{cases} \text{ as } n, d \to \infty, \frac{d}{n} \to \gamma. $$

**Approximation error** The third and fourth terms in (A.2) reflect the approximation error of $\widehat{\boldsymbol{\beta}}_{\text{OLS}}$ and $\widehat{\boldsymbol{\beta}}_{\min}$ because the linear model is misspecified for the mixture of two groups. Unlike the inductive bias and variance terms, this term does not appear in prior studies of the average/overall risk of the ridgeless least squares estimator Hastie et al. (2019).

**Lemma A.5.** *In addition to Assumption A.1, assume $\|\delta\|_2^2 = r$ and $\delta^\top \beta_0 = c$ for all $n, d$. As $d \to \infty$ we have*

$$ \delta^\top \Big\{ \begin{bmatrix} 0_{n_0 \times d} \\ X_1 \end{bmatrix}^\top X/n \Big\} (\widehat{\boldsymbol{\Sigma}}^\dagger)^2 \Big\{ X^\top \begin{bmatrix} 0_{n_0 \times d} \\ X_1 \end{bmatrix} /n \Big\} \delta \xrightarrow{p} \begin{cases} r\frac{\pi\gamma}{1-\gamma} + r\frac{\pi^2(1-2\gamma)}{1-\gamma} & \gamma < 1 \\ r\frac{\pi}{\gamma-1} + r\frac{\pi^2(\gamma-2)}{\gamma(\gamma-1)} & \gamma > 1 \end{cases} $$

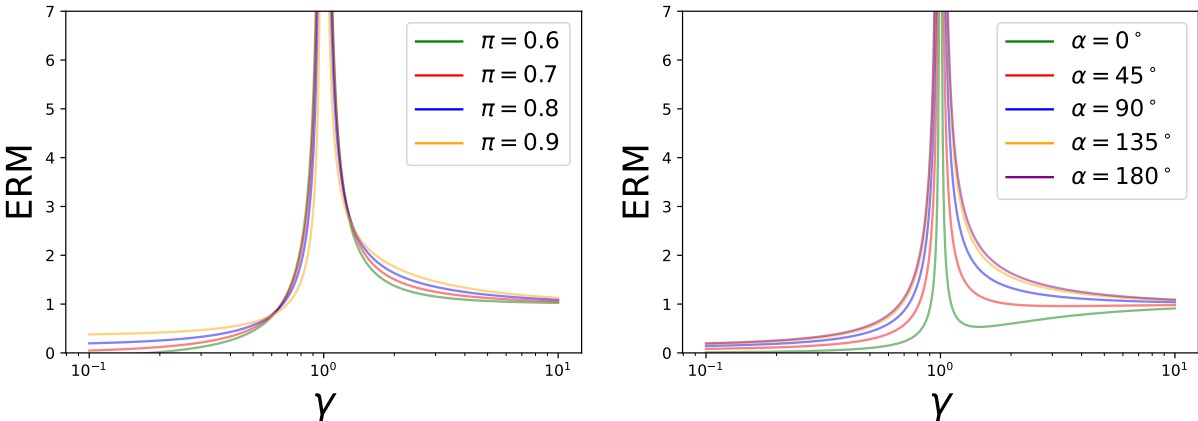

Figure 4: Minority group prediction error for the ridgeless least square ERM in the isotropic features case. The *left plot* considers the setup with varying $\pi$ when the angle between $\beta_0$ and $\beta_1$ is set at $\alpha = 180°$. The right plot sets $\pi = 0.8$ and considers different values of $\alpha$. Here the SNR is set at $\|\beta_0\|_2^2/\tau^2 = \|\beta_1\|_2^2/\tau^2 = 10$.

*and*

$$\delta^\top \left\{ \begin{bmatrix} 0_{n_0 \times d} \\ X_1 \end{bmatrix}^\top X/n \right\} (\widehat{\mathbf{\Sigma}}^\dagger)^2 \widehat{\mathbf{\Sigma}} \beta_0 \xrightarrow{p} c\pi(\{1/\gamma\} \wedge 1).$$

Before moving on, we note that (the limit of) the approximation error term is the only term that depends on the majority fraction $\pi$. Though the inductive bias may increase at overparameterized regime ($\gamma > 1$) with growing $\gamma$ when the SNR ($= \|\beta_0\|_2^2/\tau^2$) is large, we notice that the approximation error terms always decrease with growing overparameterization. In fact they tend to zero as $\gamma \to \infty$.

We plot the (the limit of) the minority risk prediction error (MSPE) in Figure 4. In both overparameterized and underparameterized regimes, the MSPE increases as $\pi$ increases. This is expected: as the fraction of training samples from the minority group decreases, we expect ERM to train a model that aligns more closely with the regression function of the majority group.

We also notice that in the overparameterized regime ($\gamma > 1$) when the SNR ($\|\beta_2\|_2^2/\tau^2$) is high the MSPE increases with growing $\gamma$ if the two groups are aligned (the angle between $\beta_0$ and $\beta_1$ is $\alpha = 0$). This trend is also observed in Hastie et al. (2019).

### A.4 Inadequacies of ridgeless least squares

There is one notable disagreement between the asymptotic risk of the ridgeless least squares estimator and the empirical practice Pham et al. (2021) in modern ML: the risk of the ridgeless least squares estimator *increases* as the overparameterization ratio $\gamma$ increases, while the accuracy of modern ML models generally improves with overparameterization. This has led ML practitioners to train overparameterized models whose risks exhibit a double-descent phenomenon Belkin et al. (2018). Inspecting the asymptotic risk of ridgeless least squares reveals the increase in risk (as $\gamma$ increases) is due to the inductive bias term (see A.3). For high SNR problems ($s_0 \gg \tau^2$), the increase of the inductive bias term dominates the decrease of the variance term (see A.4), which leads the overall risk to increase. This is a consequence of the fact that problem dimension (the dimension of the inputs) and the degree of overparameterization are tied to ridgeless least squares. In order to elucidate behavior (in the risk) that more closely matches empirical observations, we study the random features models, which allow us to keep the problem dimension fixed while increasing the overparameterization by increasing the number of random features.

### A.5 Majority group subsampling

We note that the (limiting) minority risk curve of majority group subsampling is the (limiting) minority risk curve of ERM after a change of variables. Indeed, it is not hard to check that discarding training samples

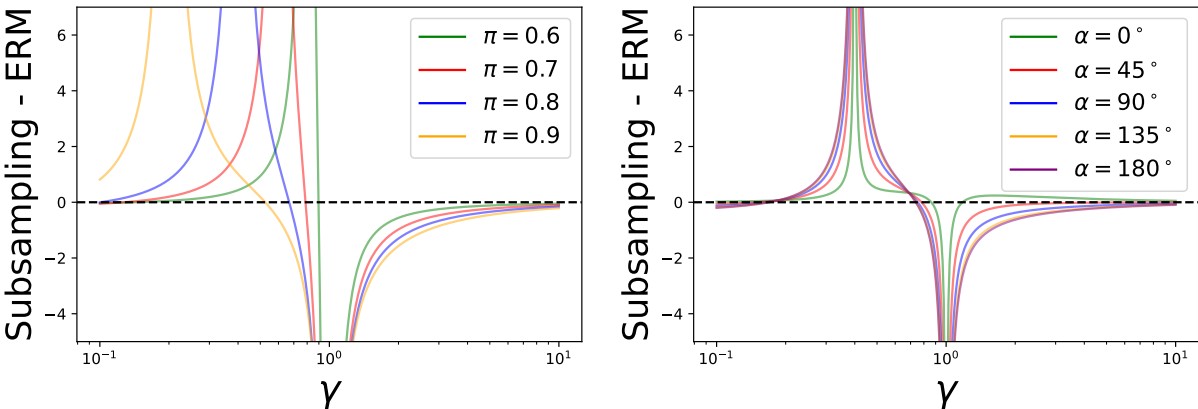

Figure 5: Minority error differnce between subsampling and ERM for the ridgeless least square in the isotropic features case. The *left plot* considers the setup with varying $\pi$ when the angle between $\beta_0$ and $\beta_1$ is set at $\alpha = 180°$. The right plot sets $\pi$ at 0.8 and consider different values of $\alpha$. Here the SNR is set at $\|\beta_0\|_2^2/\tau^2 = \|\beta_1\|_2^2/\tau^2 = 5$.

from the majority group until the groups are balanced in the training data leads to a reduction in total sample size by a factor of $2(1-\pi)$ (recall $\pi \in [\frac{1}{2}, 1]$). In other words, if the (group imbalanced) training data consists of $n$ samples, then the (group balanced) training data will have $2(1-\pi)$ training samples. In the $n, d \to \infty$, $\frac{d}{n} \to \gamma$ limit, this is equivalent to increasing $\gamma$ by a factor of $\frac{1}{2(1-\pi)}$. The limiting results in ERM can be reused with the following changes: under subsampling (1) $\pi$ changes to $1/2$, (2) $\gamma$ changes to $\gamma/(2-2\pi)$, and (3) the rest of the parameters for the limit remain the same.

In Figure 5, where we plot the differences between subsampling and ERM minority risk, we observe that subsampling generally improves minority group performance over ERM. This aligns with the empirical findings in Sagawa et al. (2020); Idrissi et al. (2021). The only instance when subsampling has worse minority risk than ERM is when the angle between $\beta_0$ and $\beta_1$ is zero, *i.e.* $\beta_0 = \beta_1$. This perfectly agrees with intuition; when there is no difference between the distributions of the two groups then subsampling discards some samples for the majority group which are valuable in learning the predictor for the minority group, resulting in an inferior predictor for the minority group.

## A.6   Proofs of Lemmas A.2 and A.5

### A.6.1   Proof of Lemma A.2

Denoting $\mathbf{U} = (X^\top X/n)^\dagger X/n$ we note that the estimation error $\widehat{\boldsymbol{\beta}} - \beta_0$ is

$$\widehat{\boldsymbol{\beta}} - \beta_0 = \mathbf{U}(X\beta_0 + \begin{bmatrix} 0_{n_0 \times d} \\ X_1 \end{bmatrix}\delta + \epsilon) - \beta_0 = (\Pi_X - I_d)\beta_0 + \mathbf{U}\begin{bmatrix} 0_{n_0 \times d} \\ X_1 \end{bmatrix}\delta + \mathbf{U}\epsilon,$$

we have the following decomposition of ERM minority risk

$$
\begin{aligned}
&R_0(\widehat{\boldsymbol{\beta}}) \\
=&\mathbb{E}\big[(\widehat{\boldsymbol{\beta}} - \beta_0)^\top(\widehat{\boldsymbol{\beta}} - \beta_0)\big] \\
=&\mathbb{E}\left[\left\{(\Pi_X - I_d)\beta_0 + \mathbf{U}\begin{bmatrix}0_{n_0 \times d}\\X_1\end{bmatrix}\delta + \mathbf{U}\epsilon\right\}^\top\left\{(\Pi_X - I_d)\beta_0 + \mathbf{U}\begin{bmatrix}0_{n_0 \times d}\\X_1\end{bmatrix}\delta + \mathbf{U}\epsilon\right\}\right] \\
=&\underbrace{\beta_0^\top(\Pi_X - I_d)^2\beta_0}_{\text{I}} + \underbrace{\mathbb{E}\big[\epsilon^\top(\mathbf{U})^\top\mathbf{U}\epsilon\big]}_{\text{II}} + \underbrace{\delta^\top\begin{bmatrix}0_{n_0 \times d}\\X_1\end{bmatrix}^\top(\mathbf{U})^\top\mathbf{U}\begin{bmatrix}0_{n_0 \times d}\\X_1\end{bmatrix}\delta}_{\text{III}} + \\
&\underbrace{2\delta^\top\begin{bmatrix}0_{n_0 \times d}\\X_1\end{bmatrix}^\top(\mathbf{U})^\top(\Pi_X - I_d)\beta_0}_{\text{IV}} + \underbrace{2\mathbb{E}\left[\left\{(\Pi_X - I_d)\beta_0 + \mathbf{U}\begin{bmatrix}0_{n_0 \times d}\\X_1\end{bmatrix}\delta\right\}^\top\mathbf{U}\epsilon\right]}_{\text{V}}.
\end{aligned}
$$

The term I is equal to $\beta_0^\top(\Pi_X - I_d)\beta_0$ because $\Pi_X - I_d$ is a projection matrix. By the linearity of expectation and trace operator, the term II is equal to

$$
\mathbb{E}\big[\text{Tr}(\epsilon^\top(\mathbf{U})^\top\mathbf{U}\epsilon)\big] = \text{Tr}\left\{\frac{1}{n}\widehat{\Sigma}^\dagger\mathbb{E}[\epsilon\epsilon^\top]\right\} = \text{Tr}\left\{\frac{1}{n}\widehat{\Sigma}^\dagger\tau^2 I_n\right\} = \frac{\tau^2}{n}\text{Tr}(\widehat{\Sigma}^\dagger).
$$

By properties of Moore–Penrose pseudoinverse, the term III and IV are equal to

$$
\delta^\top\left\{\begin{bmatrix}0_{n_0 \times d}\\X_1\end{bmatrix}^\top X/n\right\}(\widehat{\Sigma}^\dagger)^2\left\{X^\top\begin{bmatrix}0_{n_0 \times d}\\X_1\end{bmatrix}/n\right\}\delta \quad \text{and} \quad 2\delta^\top\left\{\begin{bmatrix}0_{n_0 \times d}\\X_1\end{bmatrix}^\top X/n\right\}(\widehat{\Sigma}^\dagger)^2\widehat{\Sigma}\beta_0
$$

respectively. The term V is 0 since $\mathbb{E}[\epsilon] = 0$. Hence we complete the proof. $\qquad\square$

### A.6.2 Proof of Lemma A.5

We first prove for the cross term that

$$
\mathbb{E}[\beta_0^\top\widehat{\Sigma}(\widehat{\Sigma}^\dagger)^2\frac{1}{n}X^\top(0, X_1^\top)^\top\delta] = \frac{\beta_0^\top\delta}{d}\mathbb{E}[\text{Tr}\{\widehat{\Sigma}(\widehat{\Sigma}^\dagger)^2\frac{1}{n}X^\top(0, X_1^\top)^\top\}]. \tag{A.3}
$$

To realize the above we notice that for any orthogonal matrix $O$ it holds

$$
\mathbb{E}[\beta_0^\top\widehat{\Sigma}(\widehat{\Sigma}^\dagger)^2\frac{1}{n}X^\top(0, X_1^\top)^\top\delta] = \mathbb{E}[\beta_0^\top O\widehat{\Sigma}_O(\widehat{\Sigma}_O^\dagger)^2\frac{1}{n}X_O^\top(0, X_{O,1}^\top)^\top O^\top\delta]
$$

where $X$ is replaced by $X_O = XO$ and the replacement in $X$ changes the covariance matrix to $\widehat{\Sigma}_O$ and it's Moore-Penrose inverse to $\widehat{\Sigma}_O^\dagger$. Noticing that $X \overset{d}{=} X_O$, $\widehat{\Sigma} \overset{d}{=} \widehat{\Sigma}_O$ and $\widehat{\Sigma}^\dagger \overset{d}{=} \widehat{\Sigma}_O^\dagger$ we conclude

$$
\mathbb{E}[\beta_0^\top O\widehat{\Sigma}_O(\widehat{\Sigma}_O^\dagger)^2\frac{1}{n}X_O^\top(0, X_{O,1}^\top)^\top O^\top\delta] = \mathbb{E}[\beta_0^\top O\widehat{\Sigma}(\widehat{\Sigma}^\dagger)^2\frac{1}{n}X^\top(0, X_1^\top)^\top O^\top\delta].
$$

We now let $O$ to be uniformly distributed over the set of all $d \times d$ orthogonal matrices (distributed according to Haar measure) which results in

$$
\begin{aligned}
\mathbb{E}[\beta_0^\top O\widehat{\Sigma}(\widehat{\Sigma}^\dagger)^2\frac{1}{n}X^\top(0, X_1^\top)^\top O^\top\delta] &= \mathbb{E}[\text{Tr}\{\widehat{\Sigma}(\widehat{\Sigma}^\dagger)^2\frac{1}{n}X^\top(0, X_1^\top)^\top O^\top\delta\beta_0^\top O\}] \\
&= \mathbb{E}[\text{Tr}\{\widehat{\Sigma}(\widehat{\Sigma}^\dagger)^2\frac{1}{n}X^\top(0, X_1^\top)^\top(\delta^\top\beta_0/d)\}].
\end{aligned}
$$

To calculate the term

$$
\frac{1}{d}\mathbb{E}[\text{Tr}\{\widehat{\Sigma}(\widehat{\Sigma}^\dagger)^2\frac{1}{n}X^\top(0, X_1^\top)^\top\}] = \frac{1}{d}\mathbb{E}[\text{Tr}\{\widehat{\Sigma}(\widehat{\Sigma}^\dagger)^2(\frac{1}{n}X_1^\top X_1)\}]
$$

we notice that

$$\frac{1}{d}\mathbb{E}[\mathrm{Tr}\{\ \widehat{\boldsymbol{\Sigma}}(\widehat{\boldsymbol{\Sigma}}^{\dagger})^2(\frac{1}{n}X_1^{\top}X_1)\}] = \frac{1}{d}\sum_{i=n_0+1}^{n}\mathbb{E}[\mathrm{Tr}\{\ \widehat{\boldsymbol{\Sigma}}(\widehat{\boldsymbol{\Sigma}}^{\dagger})^2(\frac{1}{n}x_ix_i^{\top})\}]$$

and each of the terms $\mathrm{Tr}\{\ \widehat{\boldsymbol{\Sigma}}(\widehat{\boldsymbol{\Sigma}}^{\dagger})^2(\frac{1}{n}x_ix_i^{\top})$ are identically distributed. We appeal to exchangability argument to conclude

$$\frac{1}{d}\sum_{i=n_0+1}^{n}\mathbb{E}[\mathrm{Tr}\{\ \widehat{\boldsymbol{\Sigma}}(\widehat{\boldsymbol{\Sigma}}^{\dagger})^2(\frac{1}{n}x_ix_i^{\top})\}] = \pi\frac{1}{d}\sum_{i=1}^{n}\mathbb{E}[\mathrm{Tr}\{\ \widehat{\boldsymbol{\Sigma}}(\widehat{\boldsymbol{\Sigma}}^{\dagger})^2(\frac{1}{n}x_ix_i^{\top})\}]$$

$$= \pi\frac{1}{d}\mathbb{E}[\mathrm{Tr}\{\ \widehat{\boldsymbol{\Sigma}}(\widehat{\boldsymbol{\Sigma}}^{\dagger})^2\widehat{\boldsymbol{\Sigma}}\}]$$

where we notice that $\widehat{\boldsymbol{\Sigma}}\widehat{\boldsymbol{\Sigma}}^{\dagger}$ is a projection matrix and conclude

$$\pi\frac{1}{d}\mathbb{E}[\mathrm{Tr}\{\ \widehat{\boldsymbol{\Sigma}}(\widehat{\boldsymbol{\Sigma}}^{\dagger})^2\widehat{\boldsymbol{\Sigma}}\}] = \pi\frac{1}{d}\mathbb{E}[\mathrm{Tr}\{\ \widehat{\boldsymbol{\Sigma}}\widehat{\boldsymbol{\Sigma}}^{\dagger}\}].$$

If $d < n$ *i.e.* $\gamma < 1$ then $\widehat{\boldsymbol{\Sigma}}$ is invertible and and it holds $\widehat{\boldsymbol{\Sigma}}\widehat{\boldsymbol{\Sigma}}^{\dagger} = \mathbb{I}_d$. If $d > n$ then there are exactly $n$ many non-zero eigen-values in $\widehat{\boldsymbol{\Sigma}}\widehat{\boldsymbol{\Sigma}}^{\dagger}$ and it holds $\mathrm{Tr}\{\ \widehat{\boldsymbol{\Sigma}}\widehat{\boldsymbol{\Sigma}}^{\dagger}\} = n$. Combining the results we obtain

$$\frac{1}{d}\mathbb{E}[\mathrm{Tr}\{\ \widehat{\boldsymbol{\Sigma}}\widehat{\boldsymbol{\Sigma}}^{\dagger}\}] = \{1/\gamma\} \wedge 1\,.$$

Now, to calculate the term

$$\mathbb{E}[\delta^{\top}\{(0, X_1^{\top})X/n\}(\widehat{\boldsymbol{\Sigma}}^{\dagger})^2\{X^{\top}(0, X_1^{\top})^{\top}\delta]$$

we first use similar argument to (A.3) and obtain

$$\mathbb{E}[\delta^{\top}\{(0, X_1^{\top})X/n\}(\widehat{\boldsymbol{\Sigma}}^{\dagger})^2\{X^{\top}(0, X_1^{\top})^{\top}\delta] = \|\delta\|_2^2\frac{1}{d}\mathbb{E}[\mathrm{Tr}\{\frac{X_1^{\top}X_1}{n}(\widehat{\boldsymbol{\Sigma}}^{\dagger})^2\frac{X_1^{\top}X_1}{n}\}]$$

We rewrite

$$\frac{1}{d}\mathbb{E}[\mathrm{Tr}\{\frac{X_1^{\top}X_1}{n}(\widehat{\boldsymbol{\Sigma}}^{\dagger})^2\frac{X_1^{\top}X_1}{n}\}] = n_1 F_{1,1} + n_1(n_1 - 1)F_{1,2}$$

where we define

$$F_{1,1} \triangleq \frac{1}{d}\mathbb{E}[\mathrm{Tr}\{\frac{x_1x_1^{\top}}{n}(\widehat{\boldsymbol{\Sigma}}^{\dagger})^2\frac{x_1x_1^{\top}}{n}\}]$$

$$F_{1,2} \triangleq \frac{1}{d}\mathbb{E}[\mathrm{Tr}\{\frac{x_1x_1^{\top}}{n}(\widehat{\boldsymbol{\Sigma}}^{\dagger})^2\frac{x_2x_2^{\top}}{n}\}]\,.$$

To calculate $nF_{1,1}$ we notice that

$$nF_{1,1} = \frac{n}{d}\mathbb{E}[\mathrm{Tr}\{\frac{x_1x_1^{\top}}{n}(\widehat{\boldsymbol{\Sigma}}^{\dagger})^2\frac{x_1x_1^{\top}}{n}\}]$$

$$= \frac{1}{\gamma}\frac{x_1^{\top}x_1}{n}\frac{x_1^{\top}(\widehat{\boldsymbol{\Sigma}}^{\dagger})^2x_1}{n}\,.$$

Denoting $\xrightarrow{L_1}$ as the $L_1$ convergence we notice that $\frac{x_1x_1^{\top}}{n} \xrightarrow{L_1} \gamma$, *i.e.* $\mathbb{E}[(\|x_1\|_2^2/n - \gamma)^2] \to 0$. We now calculate the convergence limit for $\frac{x_1^{\top}(\widehat{\boldsymbol{\Sigma}}^{\dagger})^2x_1}{n}$. Noticing that

$$\frac{1}{n}x_1^{\top}(\widehat{\boldsymbol{\Sigma}}^{\dagger})^2x_1 = \lim_{z\to 0+}\frac{1}{n}x_1^{\top}(\widehat{\boldsymbol{\Sigma}} + z\mathbb{I}_d)^{-2}x_1$$

we write $\widehat{\boldsymbol{\Sigma}} + z\mathbb{I}_d = (x_1x_1^{\top}/n) + A_z$, where $A_z = X_{-1}^{\top}X_{-1}/n + z\mathbb{I}_d$. Using the Woodbery decomposition

$$\{(x_1x_1^{\top}/n) + A_z\}^{-1} = A_z^{-1} - \frac{A_z^{-1}(x_1x_1^{\top}/n)A_z^{-1}}{1 + x_1^{\top}A_z^{-1}x_1}$$

we obtain that

$$\frac{1}{n}x_1^\top(\widehat{\boldsymbol{\Sigma}}+z\mathbb{I}_d)^{-2}x_1 = \frac{\frac{1}{n}x_1^\top A_z^{-2}x_1}{(1+\frac{1}{n}x_1^\top A_z^{-1}x_1)^2}$$

In Lemma A.6 we notice that

$$\mathbb{E}[\frac{1}{n}x_1^\top A_z^{-1}x_1] \to \begin{cases} \frac{\gamma}{1-\gamma} & \gamma < 1 \\ \frac{1}{\gamma-1} & \gamma > 1 \end{cases}, \ \mathbb{E}[\frac{1}{n}x_1^\top A_z^{-2}x_1] \to \begin{cases} \frac{\gamma}{(1-\gamma)^3} & \gamma < 1 \\ \frac{\gamma}{(\gamma-1)^3} & \gamma > 1 \end{cases} \ \text{as } d \to \infty.$$

$$\mathrm{var}[\frac{1}{n}x_1^\top A_z^{-1}x_1], \mathrm{var}[\frac{1}{n}x_1^\top A_z^{-2}x_1] \to 0$$

which implies $nF_{1,1}$ converges to

$$nF_{1,1} \xrightarrow{L_1} \begin{cases} \frac{\gamma}{1-\gamma} & \gamma < 1 \\ \frac{1}{\gamma(\gamma-1)} & \gamma > 1 \end{cases}$$

Noticing that

$$\frac{1}{d}\mathbb{E}[\mathrm{Tr}\{\ \widehat{\boldsymbol{\Sigma}}\widehat{\boldsymbol{\Sigma}}^\dagger\}] = \frac{1}{d}\mathbb{E}[\mathrm{Tr}\{\ \widehat{\boldsymbol{\Sigma}}(\widehat{\boldsymbol{\Sigma}}^\dagger)^2\widehat{\boldsymbol{\Sigma}}\}] = nF_{1,1} + n(n-1)F_{1,2}$$

we obtain the convergence of $n(n-1)F_{1,2}$ as

$$n(n-1)F_{1,2} \xrightarrow{L_1} \begin{cases} \frac{1-2\gamma}{1-\gamma} & \gamma < 1 \\ \frac{\gamma-2}{\gamma(\gamma-1)} & \gamma > 1 \end{cases}$$

which finally yields

$$\frac{1}{d}\mathbb{E}[\mathrm{Tr}\{\frac{X_1^\top X_1}{n}(\widehat{\boldsymbol{\Sigma}}^\dagger)^2\frac{X_1^\top X_1}{n}\}] = n_1 F_{1,1} + n_1(n_1-1)F_{1,2}$$

$$\asymp \pi n F_{1,1} + \pi^2 n(n-1)F_{1,2}$$

$$\xrightarrow{L_1} \begin{cases} \frac{\pi\gamma}{1-\gamma} + \frac{\pi^2(1-2\gamma)}{1-\gamma} & \gamma < 1 \\ \frac{\pi}{\gamma-1} + \frac{\pi^2(\gamma-2)}{\gamma(\gamma-1)} & \gamma > 1 \end{cases}$$

**Lemma A.6.** *As $z \to 0+$ and $d \to \infty$ we have*

$$\mathbb{E}[\frac{1}{n}x_1^\top A_z^{-1}x_1] \to \begin{cases} \frac{\gamma}{1-\gamma} & \gamma < 1 \\ \frac{1}{\gamma-1} & \gamma > 1 \end{cases},$$

$$\mathbb{E}[\frac{1}{n}x_1^\top A_z^{-2}x_1] \to \begin{cases} \frac{\gamma}{(1-\gamma)^3} & \gamma < 1 \\ \frac{\gamma}{(\gamma-1)^3} & \gamma > 1 \end{cases}$$

*and*

$$var[\frac{1}{n}x_1^\top A_z^{-1}x_1], var[\frac{1}{n}x_1^\top A_z^{-2}x_1] \to 0$$

*Proof of Lemma A.6.* To prove the results about variances we first establish that for any symmetric matrix $B = ((b_{ij})) \in \mathbb{R}^{d \times d}$ it holds

$$\mathrm{var}(x_1^\top B x_1) \leq c\mathrm{Tr}\{B^2\}, \tag{A.4}$$

for some $c > 0$. Writing $x_1 = (u_1, \ldots, u_d)^\top$ we see that

$$\mathrm{var}(x_1^\top B x_1) = \mathrm{var}\left(\sum_{i,j} u_i u_j b_{ij}\right)$$

$$= \sum_{i,j,k,l} b_{ij} b_{kl} \mathrm{cov}(u_i u_j, u_k u_l)$$

Noticing that

$$\mathrm{cov}(u_i u_j, u_k u_l) = \mathbb{E}[u_i u_j u_k u_l] - \mathbb{E}[u_i u_j]\mathbb{E}[u_k u_l]$$

we see that

1. If one of $i, j, k, l$ is distinct from the others then $\mathrm{cov}(u_i u_j, u_k u_l) = 0$.

2. If $i = j$ and $k = l$ and $i \neq k$ then

$$\mathbb{E}[u_i u_j u_k u_l] - \mathbb{E}[u_i u_j]\mathbb{E}[u_k u_l] = \mathbb{E}[u_i^2 u_k^2] - \mathbb{E}[u_i^2]\mathbb{E}[u_k^2] = 0\,.$$

3. If $i = j = k = l$ then $\mathrm{cov}(u_i u_j, u_k u_l) = \mathrm{var}(u_1^2)$.

4. If $\{i = k \text{ and } j = l \text{ and } i \neq j\}$ or $\{i = l \text{ and } j = k \text{ and } i \neq j\}$ then we have

$$\mathbb{E}[u_i u_j u_k u_l] - \mathbb{E}[u_i u_j]\mathbb{E}[u_k u_l] = \mathrm{var}(u_i u_j) = \mathrm{var}(u_1 u_2)\,.$$

Gathering the terms we notice that

$$\mathrm{var}(x_1^\top B x_1) = \sum_i \mathrm{var}(u_i^2) b_{ii}^2 + 2 \sum_{i \neq j} b_{ij}^2 \mathrm{var}(u_i u_j)$$
$$\leq c \sum_{i,j} b_{ij}^2 = c\mathrm{Tr}\{B^2\}\,,$$

where $c = 2\{\mathrm{var}(u_1^2) \vee \mathrm{var}(u_1 u_2)\}$.

Using (A.4) we notice that

$$\lim_{z \to 0+} \mathrm{var}(x_1^\top A_z^{-j} x_1/n) = \frac{1}{n^2}\mathrm{Tr}\{(\widehat{\boldsymbol{\Sigma}}_{-1}^\dagger)^{2j}\} \asymp \frac{1}{n^2}\mathrm{Tr}\{(\widehat{\boldsymbol{\Sigma}}^\dagger)^{2j}\} \to 0\,.$$

We notice that

$$\mathbb{E}[\frac{1}{n}x_1^\top A_z^{-1} x_1] = \frac{1}{n}\mathrm{Tr}\{A_z^{-1}\} \asymp \frac{1}{n}\mathrm{Tr}\{\widehat{\boldsymbol{\Sigma}}^\dagger\}$$

which appears in the variance term in Lemma A.4, and we conclude that

$$\mathbb{E}[\frac{1}{n}x_1^\top A_z^{-1} x_1] \to \begin{cases} \frac{\gamma}{1-\gamma} & \gamma < 1 \\ \frac{1}{\gamma-1} & \gamma > 1 \end{cases} \quad \text{as } d \to \infty.$$

To calculate

$$\mathbb{E}[\frac{1}{n}x_1^\top A_z^{-2} x_1] = \frac{1}{n}\mathrm{Tr}\{A_z^{-2}\} \asymp \frac{1}{n}\mathrm{Tr}\{(\widehat{\boldsymbol{\Sigma}}^\dagger)^2\}$$

we consider case by case.

For $\gamma < 1$ we see that

$$\frac{1}{n}\mathrm{Tr}\{(\widehat{\boldsymbol{\Sigma}}^\dagger)^2\} = \gamma\frac{1}{d}\mathrm{Tr}\{(\widehat{\boldsymbol{\Sigma}})^{-2}\}$$
$$\to \gamma\int \frac{1}{s^2} dF_\gamma(s)$$

The Stieltjes transformation of $\mu_\gamma$ is

$$s_\gamma(z) = \int \frac{1}{x-z} d\mu_\gamma(x)$$
$$= \frac{1-\gamma-z-\sqrt{(1-\gamma-z)^2-4\gamma z}}{2\gamma z}, \quad z \in \mathbb{C}\backslash[1-\sqrt{\gamma}, 1+\sqrt{\gamma}].$$

For $|z| < 1 - \sqrt{\gamma}$

$$s_\gamma(z) = \int \frac{1}{x-z} d\mu_\gamma(x)$$
$$= \int \frac{1}{x}\frac{1}{1-z/x} d\mu_\gamma(x)$$
$$= \int \frac{1}{x}\sum_{k=0}^{\infty}(z/x)^k d\mu_\gamma(x)$$

Hence, we have

$$\lim_{z \to 0-} \frac{d}{dz}\big(s_\gamma(z)\big) = \mathbb{E}[1/X^2].$$

Now,

$$1 - \gamma - z - \sqrt{(1 - \gamma - z)^2 - 4\gamma z}$$

$$= 1 - \gamma - z - \big(1 + \gamma^2 + z^2 - 2\gamma - 2z + 2\gamma z - 4\gamma z\big)^{1/2}$$

$$= 1 - \gamma - z - \big((1 - \gamma)^2 + z^2 - 2z - 2\gamma z\big)^{1/2}$$

$$\asymp 1 - \gamma - z - (1 - \gamma)\Big[1 + \frac{z^2 - 2z - 2\gamma z}{2(1 - \gamma)^2} - \frac{(z^2 - 2z - 2\gamma z)^2}{8(1 - \gamma)^4}\Big], \quad \text{as } z \to 0-$$

$$\asymp 1 - \gamma - z - (1 - \gamma)\Big[1 + \frac{z^2 - 2z - 2\gamma z}{2(1 - \gamma)^2} - \frac{z^2(1 + \gamma)^2}{2(1 - \gamma)^4}\Big], \quad \text{as } z \to 0-$$

$$= \frac{z^2(1 + \gamma)^2}{2(1 - \gamma)^3} - \frac{z^2 - 4\gamma z}{2(1 - \gamma)}$$

$$= \frac{2z^2\gamma}{(1 - \gamma)^3} + \frac{2\gamma z}{1 - \gamma}$$

which implies

$$s_\gamma(z) \asymp \frac{z}{(1 - \gamma)^3} + \frac{1}{1 - \gamma}$$

This establish that

$$\frac{1}{n}\text{Tr}\{(\widehat{\boldsymbol{\Sigma}}^\dagger)^2\} = \gamma \lim_{z \to 0-} \frac{d}{dz}\big(s_\gamma(z)\big) = \frac{\gamma}{(1 - \gamma)^3} \,.$$

For $\gamma > 1$ we notice that

$$\frac{1}{n}\text{Tr}\{(\widehat{\boldsymbol{\Sigma}}^\dagger)^2\} = \frac{1}{n}\sum_{i=1}^{n} \frac{1}{s_i^2}$$

where $s_i$'s are non-zero eigen-values of $X^\top X/n$. This is also equal to

$$\frac{1}{\gamma p}\sum_{i=1}^{n} \frac{1}{t_i^2}$$

where $t_i$ are the eigen-values of the invertible matrix $XX^\top/d$. Hence we have

$$\frac{1}{n}\text{Tr}\{(\widehat{\boldsymbol{\Sigma}}^\dagger)^2\} = \frac{1}{\gamma^2}\frac{1}{n}\text{Tr}\{(XX^\top/d)^{-2}\} \to \frac{1}{\gamma^2}\lim_{z \to 0+} \frac{d}{dz}\big(s_{1/\gamma}(z)\big) = \frac{\gamma}{(\gamma - 1)^3} \,.$$

We combine the limits for $\gamma < 1$ and $\gamma > 1$ to write to write

$$\mathbb{E}[\frac{1}{n}x_1^\top A_z^{-2}x_1] \to \begin{cases} \frac{\gamma}{(1-\gamma)^3} & \gamma < 1 \\ \frac{\gamma}{(\gamma-1)^3} & \gamma > 1 \end{cases} \quad \text{as } d \to \infty.$$

$\square$

## B  Proof of Theorem 3.9

Before diving into the main proof we give a rough sketch of the whole proof. Essentially, the proof has three major components.

1. In the first part, we establish the limiting risk where we treat $\beta_0$ and $\delta$ as uncorrelated random variables. (Section B.1)

2. In the second part of the proof we establish the necessary bias-variance decomposition under orthogonality of the fixed parameters $\beta_0$ and $\delta$. (Section B.2)

3. In the final part, we use the results of previous two parts to establish the limiting risk result for $\beta_0$ and $\delta$ in general position. (Section B.3 and B.4)

### B.1 Limiting risk for uncorrelated parameters

We start this section with some definitions of some important quantities.

**Definition B.1.** *Let the functions $\nu_1, \nu_2 : \mathbb{C}_+ \to \mathbb{C}_+$ be uniquely defined by the following conditions: (i) $\nu_1, \nu_2$ are analytic on $\mathbb{C}_+$. (ii) For $Im(\zeta) > 0$, $\nu_1(\zeta)$ and $\nu_2(\zeta)$ satisfy the equations*

$$\nu_1 = \psi_1 \left( -\zeta - \nu_2 - \frac{\xi^2 \nu_2}{1 - \xi^2 \nu_1 \nu_2} \right)^{-1}$$

$$\nu_2 = \psi_1 \left( -\zeta - \nu_1 - \frac{\xi^2 \nu_1}{1 - \xi^2 \nu_1 \nu_2} \right)^{-1}$$

*(iii) $(\nu_1(\zeta), \nu_2(\zeta))$ is the unique solution of these equations with*

$$|\nu_1(\zeta)| \le \psi_1/Im(\zeta), \ |\nu_2(\zeta)| \le \psi_2/Im(\zeta) \ for \ Im(\zeta) > C,$$

*with a $C$ sufficiently large constant.*

*Let*

$$\chi := \nu_1(i(\psi_1 \psi_2 \lambda)^{1/2}).\nu_2(i(\psi_1 \psi_2 \lambda)^{1/2}),$$

*and*

$$
\begin{aligned}
\mathscr{E}_0(\xi, \psi_1, \psi_2, \lambda) = {}& -\chi^5 \xi^6 + 3\chi^4 \xi^4 + (\psi_1 \psi_2 - \psi_1 - \psi_2 + 1)\chi^3 \xi^6 \\
& - 2\chi^3 \xi^4 - 3\chi^3 \xi^2 + (\psi_1 + \psi_2 - 3\psi_1 \psi_2 + 1)\chi^2 \xi^4 + 2\chi^2 \xi^2 \\
& + \chi^2 + 3\psi_1 \psi_2 \chi \xi^2 - \psi_1 \psi_2, \\
\mathscr{E}_1(\xi, \psi_1, \psi_2, \lambda) = {}& \psi_2 \chi^3 \xi^4 - \psi_2 \chi^2 \xi^2 + \psi_1 \psi_2 \chi \xi^2 - \psi_1 \psi_2, \\
\mathscr{E}_2(\xi, \psi_1, \psi_2, \lambda) = {}& \chi^5 \xi^6 - 3\chi^4 \xi^4 + (\psi_1 - 1)\chi^3 \xi^6 + 2\chi^3 \xi^4 + 3\chi^3 \xi^2 \\
& + (-\psi_1 - 1)\chi^2 \xi^4 - 2\chi^2 \xi^2 - \chi^2.
\end{aligned}
$$

*We then define*

$$\mathscr{B}_{\text{ridge}}(\xi, \psi_1, \psi_2, \lambda) = \frac{\mathscr{E}_1(\xi, \psi_1, \psi_2, \lambda)}{\mathscr{E}_0(\xi, \psi_1, \psi_2, \lambda)}, \tag{B.1}$$

$$\mathscr{V}_{\text{ridge}}(\xi, \psi_1, \psi_2, \lambda) = \frac{\mathscr{E}_2(\xi, \psi_1, \psi_2, \lambda)}{\mathscr{E}_0(\xi, \psi_1, \psi_2, \lambda)}. \tag{B.2}$$

The quantities derived above can be easily derived numerically. But for our interest, we need to focus on the limiting case when $\lambda \to 0$. It can be shown that when $\lambda \to 0$, the expressions of $\mathscr{B}_{\text{ridge}}$ and $\mathscr{V}_{\text{ridge}}$ reduces to the form of $\mathscr{B}^\star$ and $\mathscr{V}^\star$ defined in Lemma 3.7 respectively. For details we refer to Section 5.2 in Mei & Montanari (2019).

Now we are ready to present the proof of the main result. Recall that

$$\hat{a}(\lambda) := \arg\min_{a \in \mathbb{R}^N} \frac{1}{n} \sum_{i=1}^n \left( y_i - \sum_{j=1}^N a_j \sigma(\theta_j^\top x_i/\sqrt{d}) \right)^2 + \frac{N\lambda}{d} \|a\|_2^2.$$

By standard linear algebra it immediately follows that

$$\hat{a}(\lambda) = \frac{1}{\sqrt{d}} (Z^\top Z + \lambda \psi_{1,d} \psi_{2,d} \mathbb{I}_N)^{-1} Z^\top y, \tag{B.3}$$

where $Z = \frac{1}{\sqrt{d}}\sigma(X\Theta^\top/\sqrt{d})$, $\psi_{1,d} = \frac{N}{d}$ and $\psi_{2,d} = \frac{n}{d}$. Also, define the square error loss $R_{\mathrm{RF}}(x, X, \Theta, \lambda) := (x^\top\beta_0 - f(x; \hat{a}(\lambda); \Theta))^2$, where $x$ is a new feature point.

For brevity, we denote by $\boldsymbol{\Gamma}$ the tuple $(X, \Theta, \beta_0, \delta)$. Recall that we are interested in the out-of-sample risk $\mathbb{E}_{\boldsymbol{\Gamma}}\mathbb{E}_x[R_{\mathrm{RF}}(x, X, \Theta, \lambda)]$, where $x$ is a new feature point from minority group. The main recipe of our proof is the following:

- We first do the bias-variance decomposition of the expected out-of-sample-risk.

- We analyze the bias and variance terms separately using techniques from random matrix theory.

- Finally, we obtain asymptotic limit of the expected risk by using the asymptotic limits of bias and variance terms.

**Bias variance decomposition**

The expected risk at a new point $x$ coming from the minority group can be decomposed into following way:

$$
\begin{aligned}
\mathbb{E}_{\boldsymbol{\Gamma}}\mathbb{E}_x[R_{\mathrm{RF}}(x, X, \Theta, \lambda)] &= \mathbb{E}_{\boldsymbol{\Gamma}}\mathbb{E}_x\{x^\top\beta_0 - f(x; \hat{a}(\lambda); \Theta)\}^2 \\
&= \underbrace{\mathbb{E}_{\boldsymbol{\Gamma}}\mathbb{E}_x\left\{\left[x^\top\beta_0 - \mathbb{E}_\epsilon f(x; \hat{a}(\lambda); \Theta)\right]^2\right\}}_{\text{Bias term}} \\
&\quad + \underbrace{\mathbb{E}_{\boldsymbol{\Gamma}}\mathbb{E}_x\mathrm{Var}_\epsilon\left\{f(x; \hat{a}(\lambda); \Theta)\right\}}_{\text{Variance term}}.
\end{aligned}
\tag{B.4}
$$

The above fact follows from Lemma 3.3, which basically uses the uncorrelatedness of $\beta_0$ and $\delta$. Now we define the matrix

$$
\Psi := (Z^\top Z + \lambda\psi_{1,d}\psi_{2,d}\mathbb{I}_N)^{-1}.
\tag{B.5}
$$

In the next couple of sections we will study the asymptotic behavior of the bias and variance terms by analyzing several terms involving $\Psi$. In order to obtain asymptotic limits of those terms, we will be borrowing results of random matrix theory in Mei & Montanari (2019).

**Decomposition of bias term**

We focus the bias

$$
\begin{aligned}
\mathcal{B}(\delta, \lambda) &:= \mathbb{E}_{\boldsymbol{\Gamma}}\mathbb{E}_x\left\{\left[x^\top\beta_0 - \mathbb{E}_\epsilon f(x; \hat{a}(\lambda); \Theta)\right]^2\right\} \\
&= \mathbb{E}_{\Gamma}\mathbb{E}_x\left\{\left[x^\top\beta_0 - \sigma(\Theta x/\sqrt{d})^\top \frac{1}{\sqrt{d}}\left(\Psi Z^\top X\beta_0 + \Psi Z_1^\top X_1\delta\right)\right]^2\right\}.
\end{aligned}
\tag{B.6}
$$

It turns out that it is rather difficult to directly analyze the bias term $\mathcal{B}(\delta, \lambda)$. The main difficulty lies in the fact that the terms involving $\beta_0$ and $\delta$ are coupled in the bias term and poses main difficulty in applying well known random matrix theory results directly. Hence, in order to decouple the terms we look at the following decomposition:

$$
\mathcal{B}(\delta, \lambda) = \underbrace{\mathcal{B}(\delta, \lambda) - \mathcal{B}(0, \lambda)}_{(I)} + \underbrace{\mathcal{B}(0, \lambda)}_{(II)}
$$

It is fairly simple to see that term (II) is only a function of $\beta_0$. Next, we will show that term (I) does not contain $\beta_0$. To this end, We define the matrices $\widehat{U}, U \in \mathbb{R}^{N \times N}$ as follows:

$$
\widehat{U}_{ij} = \sigma(\theta_i^\top x/\sqrt{d})\sigma(\theta_j^\top x/\sqrt{d}), \ U_{ij} = \mathbb{E}(\widehat{U}_{ij}) \ \forall(i, j) \in [N] \times [N].
\tag{B.7}
$$

Next, due to Assumption 3.2, it easily follows that

$$
\begin{aligned}
&\mathcal{B}(\delta, \lambda) - \mathcal{B}(0, \lambda) \\
&= \mathbb{E}_{\boldsymbol{\Gamma}}\mathbb{E}_x\left\{\left[x^\top\beta_0 - \sigma(\Theta x/\sqrt{d})^\top\frac{1}{\sqrt{d}}\big(\Psi Z^\top X\beta_0 + \Psi Z_1^\top X_1\delta\big)\right]^2\right\} \\
&\quad - \mathbb{E}_{\boldsymbol{\Gamma}}\mathbb{E}_x\left\{\left[x^\top\beta_0 - \sigma(\Theta x/\sqrt{d})^\top\frac{1}{\sqrt{d}}\big(\Psi Z^\top X\beta_0\big)\right]^2\right\} \\
&= \frac{1}{d}\mathbb{E}_{\boldsymbol{\Gamma}}\mathbb{E}_x\left(\delta^\top X_1^\top Z_1\Psi\widehat{U}\Psi Z_1^\top X_1\delta\right) \\
&= \frac{1}{d}\mathbb{E}_{\boldsymbol{\Gamma}}\mathbb{E}_x\left\{\mathrm{Tr}\big(X_1^\top Z_1\Psi\widehat{U}\Psi Z_1^\top X_1\delta\delta^\top\big)\right\} \\
&= \frac{F_\delta^2}{d^2}\mathbb{E}\left\{\mathrm{Tr}\left(X_1^\top Z_1\Psi U\Psi Z_1^\top X_1\right)\right\}.
\end{aligned}
\tag{B.8}
$$

Above display shows that term (I) does not contain $\beta_0$. In the subsequent discussion, we will now focus on obtaining the limits of term (I) and (II) separately.

Next, by the property of trace, we have the following:

$$
\begin{aligned}
\mathrm{Tr}(X_1^\top Z_1\Psi U\Psi Z_1^\top X_1) &= \mathrm{Tr}\left\{\left(\sum_{i=n_0+1}^n x_i z_i^\top\right)\Psi U\Psi\left(\sum_{i=n_0+1}^n z_i x_i^\top\right)\right\} \\
&= \sum_{i=n_0+1}^{n_1}\sum_{j=n_0+1}^{n_1}\mathrm{Tr}\left(\Psi U\Psi z_j x_j^\top x_i z_i^\top\right) \\
&= \sum_{i=n_0+1}^{n_1}\sum_{j=n_0+1}^{n_1} x_j^\top x_i\mathrm{Tr}\left(\Psi U\Psi z_j z_i^\top\right) \\
&= \sum_{i=n_0+1}^{n} x_i^\top x_i\mathrm{Tr}\left(\Psi U\Psi z_i z_i^\top\right) \\
&\quad + \sum_{n_0+1\le i<j\le n} x_j^\top x_i\mathrm{Tr}\left(\Psi U\Psi z_j z_i^\top\right).
\end{aligned}
\tag{B.9}
$$

Thus, by exchangeability of the terms $\{\mathrm{Tr}\left(\Psi U\Psi z_i z_i^\top\right)\}_{i\in[n]}$ and $\{x_j^\top x_i\mathrm{Tr}\left(\Psi U\Psi z_j z_i^\top\right)\}_{n_0+1\le i<j\le n}$ we have

$$
\frac{1}{d}\mathbb{E}\left\{\mathrm{Tr}\left(X_1^\top Z_1\Psi U\Psi Z_1^\top X_1\right)\right\} = n_1 f(1,1) + n_1(n_1-1)f(1,2),
\tag{B.10}
$$

where $f(i,j) = \mathbb{E}\left\{x_j^\top x_i\mathrm{Tr}\left(\Psi U\Psi z_j z_i^\top\right)\right\}/d^2$ and $1\le i,j\le 2$. Hence, together with Equation (B.8) and Equation (B.10), it follows that

$$
\mathcal{B}(\delta, \lambda) - \mathcal{B}(0, \lambda) = F_\delta^2\{n_1 f(1,1) + n_1(n_1-1)f(1,2)\}.
\tag{B.11}
$$

Furthermore, from Equation (8.10), Equation (8.25), Lemma 9.3 and Lemma 9.4 of Mei & Montanari (2019), we get

$$
\mathcal{B}(0, \lambda) = \mathscr{B}_{\mathrm{ridge}}(\xi, \psi_1, \psi_2, \lambda/\mu_\star^2)F_\beta^2 + o_d(1),
\tag{B.12}
$$

where $\mathscr{B}_{\mathrm{ridge}}(\xi, \psi_1, \psi_2, \lambda/\mu_\star^2)$ is defined as in Equation (B.1). Thus, it only remains to calculate the limit of $\mathcal{B}(\delta, \lambda) - \mathcal{B}(0, \lambda)$ in order to obtain the limit of the bias term $\mathcal{B}(\delta, \lambda)$.

**Analysis of variance term**

Now we briefly deffer the calculation of term (I) in Equation (B.11) and shift our focus to variance term. The reason behind this is that it is hard to directly calculate the limits of the quantities appearing in the right hand side of Equation (B.11). We obtain these limiting quantities with the help of the limiting variance.

Note that for the variance term, we have

$$
\begin{aligned}
\mathcal{V}(\lambda) &= \mathbb{E}_{\boldsymbol{\Gamma}}\mathbb{E}_x \text{Var}_\epsilon\big\{f(x;\hat{a}(\lambda);\Theta)\big\} \\
&= \mathbb{E}_{\boldsymbol{\Gamma}}\mathbb{E}_x \text{Var}_\epsilon\Big\{\sigma(\Theta x/\sqrt{d})^\top \frac{1}{\sqrt{d}}\Psi Z^\top \epsilon\Big\} \\
&= \frac{1}{d}\mathbb{E}_{\boldsymbol{\Gamma}}\mathbb{E}_x\Big[\text{Tr}\big\{\Psi\widehat{U}\Psi Z^\top Z\big\}\Big]\tau^2 \\
&= \frac{\tau^2}{d}\mathbb{E}\big\{\text{Tr}\big(\Psi U\Psi Z^\top Z\big)\big\} \\
&= \widetilde{\Psi}_3 \tau^2,
\end{aligned}
\tag{B.13}
$$

where $\widetilde{\Psi}_3 := \mathbb{E}\big\{\text{Tr}\big(\Psi U\Psi Z^\top Z\big)\big\}/d$. Again by exchangeability argument, we have the following:

$$
\begin{aligned}
nf(1,1) &= \mathbb{E}\Big\{\frac{1}{d^2}\sum_{i=1}^n x_i^\top x_i \text{Tr}\big(\Psi U\Psi z_i z_i^\top\big)\Big\} \\
&= \mathbb{E}\Big\{\frac{1}{d}\sum_{i=1}^n \text{Tr}\big(\Psi U\Psi z_i z_i^\top\big)\Big\} \qquad (\text{as } x_i^\top x_i/d = 1) \\
&= \mathbb{E}\Big\{\frac{1}{d}\text{Tr}\big(\Psi U\Psi Z^\top Z\big)\Big\} \\
&= \widetilde{\Psi}_3.
\end{aligned}
\tag{B.14}
$$

Thus, in the light of Equation (B.11) and Equation (B.14), we reemphasize that it is essential to understand the limiting behavior of $f(i,j)$ in order to obtain the limit of expected out-of-sample risk.

**Obtaining limit of variance term:**

To this end, we define the matrix $\mathbf{A} \in \mathbb{R}^{M \times M}, M = N + n$ with parameters $\mathbf{q} = (s_1, s_2, t_1, t_2, p) \in \mathbb{R}^5$ in the following way:

$$
\mathbf{A} = \mathbf{A}(\mathbf{q}) := \begin{bmatrix} s_1\mathbf{I}_N + s_2\mathbf{Q} & Z^\top + p\mathbf{W}^\top \\ Z + p\mathbf{W} & t_1\mathbf{I}_n + t_2\mathbf{H} \end{bmatrix}
$$

where

$$
\mathbf{Q} = \frac{1}{d}\Theta\Theta^\top, \mathbf{H} = \frac{1}{d}XX^\top, \mathbf{W} = \frac{\mu_1}{d}X\Theta^\top.
$$

For $\xi \in \mathbb{C}_+$, we define the log-determinant of $\mathbf{A}$ as
$G(\xi, \mathbf{q}) = \Big[\frac{1}{d}\sum_{i=1}^M \text{Log}\big(\lambda_i(\mathbf{A}) - \xi\mathbf{I}_M\big)\Big]$. Here Log is the branch cut on the negative real axis and $\{\lambda_i(\mathbf{A})\}_{i\in[M]}$ denotes the eigenvalues of $\mathbf{A}$ in non-increasing order.

Define the quantity

$$
\breve{\Psi}_3 := \frac{1}{d}\text{Tr}\big(\Psi U\Psi Z^\top Z\big).
$$

From Equation (B.14), it trivially follows that $\mathbb{E}(\breve{\Psi}_3) = nf(1,1)$. The key trick lies in replacing the kernel matrix $U$ by the matrix $\boldsymbol{\Lambda} = \mu_1^2\mathbf{Q} + \mu_\star^2\mathbf{I}_N$. By (Mei & Montanari, 2019, Lemma 9.4), we know that the asymptotic error incurred in the expected value of $\breve{\Psi}_3$ by replacing $U$ in place of $\boldsymbol{\Lambda}$ is $o_d(1)$. To elaborate, we first define

$$
\Psi_3 := \frac{1}{d}\text{Tr}\big(\Psi\boldsymbol{\Lambda}\Psi Z^\top Z\big).
$$

Then by (Mei & Montanari, 2019, Lemma 9.4), we have

$$
\mathbb{E}|\breve{\Psi}_3 - \Psi_3| = o_d(1).
\tag{B.15}
$$

Thus, instead of $\breve{\Psi}_3$, we will focus on $\Psi_3$. By (Mei & Montanari, 2019, Proposition 8.2) we know

$$\Psi_3 = -\mu_\star^2 \partial_{s_1,t_1} G_d(i(\psi_1\psi_2\lambda)^{1/2}; \mathbf{0}) - \mu_1^2 \partial_{s_2,t_1} G_d(i(\psi_1\psi_2\lambda)^{1/2}; \mathbf{0}). \tag{B.16}$$

Also by (Mei & Montanari, 2019, Proposition 8.5), for any fixed $u \in \mathbb{R}_+$, we have the following:

$$\lim_{d\to\infty} \mathbb{E}\left[\left\|\nabla_{\mathbf{q}}^2 G_d(iu; \mathbf{0}) - \nabla_{\mathbf{q}}^2 g(iu; \mathbf{0})\right\|_{op}\right] = 0. \tag{B.17}$$

As $\Psi_3$ is a bi-linear form of $\nabla_{\mathbf{q}}^2 G_d(i(\psi_1\psi_2\lambda)^{1/2}; \mathbf{0})$ (See Equation (B.16)), together with (Mei & Montanari, 2019, Equation (8.26)), it follows that

$$\lim_{d\to\infty} \mathbb{E}\left|\Psi_3 - \mathscr{V}_{\mathrm{ridge}}(\xi, \psi_1, \psi_2, \lambda/\mu_\star^2)\right| \to 0, \tag{B.18}$$

where $\mathscr{V}_{\mathrm{ridge}}(\xi, \psi_1, \psi_2, \lambda/\mu_\star^2)$ is as defined in Equation (B.2). Thus, both Equation (B.15) and Equation (B.18) yields that

$$\lim_{d\to\infty} \mathbb{E}|\breve{\Psi}_3 - \mathscr{V}_{\mathrm{ridge}}(\xi, \psi_1, \psi_2, \lambda/\mu_\star^2)| \to 0. \tag{B.19}$$

Hence, by Equation (B.13) we get the following:

$$\mathcal{V}(\lambda) = \mathscr{V}_{\mathrm{ridge}}(\xi, \psi_1, \psi_2, \lambda/\mu_\star^2)\tau^2 + o_d(1). \tag{B.20}$$

**Obtaining limit of bias term:**

Now we revisit the term $\mathcal{B}(\delta, \lambda) - \mathcal{B}(0, \lambda)$. We will study the terms $n_1 f(1,1)$ and $n_1(n_1-1)f(1,2)$ in Equation (B.11) separately.

First, we focus on the term $n_1 f(1,1)$. Equation (B.14), Equation (B.15) and Equation (B.19) show that

$$nf(1,1) = \mathscr{V}_{\mathrm{ridge}}(\xi, \psi_1, \psi_2, \lambda/\mu_\star^2) + o_d(1). \tag{B.21}$$

Next, in order to understand the limiting behavior of $f(1,2)$, we define

$$\breve{\Psi}_2 := \frac{1}{d}\mathrm{Tr}\left(\Psi U \Psi Z^\top \mathbf{H} Z\right), \quad \Psi_2 := \frac{1}{d}\mathrm{Tr}\left(\Psi \mathbf{\Lambda} \Psi Z^\top \mathbf{H} Z\right).$$

Again due to (Mei & Montanari, 2019, Lemma 9.4), we have

$$\mathbb{E}|\breve{\Psi}_3 - \Psi_3| = o_d(1).$$

Hence, we only focus on $\Psi_2$. A calculation similar to (B.9) shows that

$$\mathbb{E}(\Psi_2) = nf(1,1) + n(n-1)f(1,2) + o_d(1). \tag{B.22}$$

Let $g(\xi; \mathbf{q})$ be the analytic function defined in (Mei & Montanari, 2019, Equation (8.19)). Using (Mei & Montanari, 2019, Proposition 8.5), we get

$$\mathbb{E}(\Psi_2) = -\mu_\star^2 \partial_{s_1,t_2} g(i(\psi_1\psi_2\lambda)^{1/2}; \mathbf{0}) - \mu_1^2 \partial_{s_2,t_2} g(i(\psi_1\psi_2\lambda)^{1/2}; \mathbf{0}) + o_d(1)$$
$$=: \Psi_2^\star(\xi, \psi_1, \psi_2, \lambda, \mu_\star, \mu_1) + o_d 1.$$

This along with (B.21) and (B.22) show that

$$n(n-1)f(1,2) = \Psi_2^\star(\xi, \psi_1, \psi_2, \lambda, \mu_\star, \mu_1) - \mathscr{V}_{\mathrm{ridge}}(\xi, \psi_1, \psi_2, \lambda/\mu_\star^2) + o_d(1). \tag{B.23}$$

Thus, finally using Equation (B.11), Equation (B.12) and the fact $n_1/n \to \pi$, we conclude that

$$\mathcal{B}(\delta, \lambda) = F_\beta^2 \mathscr{B}_{\mathrm{ridge}}(\xi, \psi_1, \psi_2, \lambda/\mu_\star^2) + F_\delta^2 [\pi(1-\pi)\mathscr{V}_{\mathrm{ridge}}(\xi, \psi_1, \psi_2, \lambda/\mu_\star^2)$$
$$+ \pi^2 \Psi_2^\star(\xi, \psi_1, \psi_2, \lambda, \mu_\star, \mu_1)] + o_d(1). \tag{B.24}$$

Lastly, using Equation (B.20), Equation (B.24) and plugging the values in Equation (B.4) we get

$$\lim_{d\to\infty} \mathbb{E}_{X,\Theta,\beta_0,\delta}[R_{\mathrm{RF}}(x, X, \Theta, \lambda)]$$
$$= F_\beta^2 \mathscr{B}_{\mathrm{ridge}}(\xi, \psi_1, \psi_2, \lambda/\mu_\star^2) + F_\delta^2 [\pi(1-\pi)\mathscr{V}_{\mathrm{ridge}}(\xi, \psi_1, \psi_2, \lambda/\mu_\star^2)$$
$$+ \pi^2 \Psi_2^\star(\xi, \psi_1, \psi_2, \lambda, \mu_\star, \mu_1)] + \tau^2 \mathscr{V}_{\mathrm{ridge}}(\xi, \psi_1, \psi_2, \lambda/\mu_\star^2). \tag{B.25}$$

**Ridgeless limit**

Finally, for ridgeless limit we need to take $\lambda \to 0+$ on both sides of Equation (B.25). Following similar calculations as in the proof of (Mei & Montanari, 2019, Theorem 5.7), or more specifically using (Mei & Montanari, 2019, Lemma 12.1), we ultimately get

$$\lim_{\lambda \to 0+} \lim_{d \to \infty} \mathbb{E}_{X,\Theta,\beta_0,\delta}[R_{\mathrm{RF}}(x, X, \Theta, \lambda)] = F_\beta^2 \mathscr{B}^\star + F_\delta^2 \mathscr{M}_1^\star + \tau^2 \mathscr{V}^\star.$$

This completes the proof.

## B.2 Bias-variance decomposition under orthogonality of parameters

In this section we will demonstrate that the same bias-variance decomposition in Lemma 3.3 continues to hold rather weaker assumption than Assumption 3.2. Specifically, in this section we only assume the parameter vectors $\beta_0$ and $\delta$ are orthogonal, i.e., $\beta_0^\top \delta = 0$. The following Lemma shows that under this orthogonality condition, the desired bias-variance decomposition still holds.

**Lemma B.2.** *Define the followings:* $\mathscr{B}_\beta = \mathbb{E}_{x \sim P_X}[\|z^\top (Z^\top Z)^\dagger Z^\top X - x^\top\|_2^2 / d]$ *and* $\mathscr{B}_\delta = \mathbb{E}_{x \sim P_X}[\|z^\top (Z^\top Z)^\dagger Z_1^\top X_1\|_2^2 / d]$. *Also assume that* $\beta_0^\top \delta = 0$. *Then, we have* $\mathscr{B}(\beta_0, \delta) = F_\beta^2 \mathscr{B}_\beta + F_\delta^2 \mathscr{B}_\delta$.

*Proof.* Similar to Equation (B.4), her also we have

$$\begin{aligned}
\mathscr{B}(\beta_0, \delta) = \mathbb{E}_{x \sim P_X}[\{(z^\top (Z^\top Z)^\dagger Z^\top X - x^\top)\beta_0\}^2 + \{z^\top (Z^\top Z)^\dagger Z_1^\top X_1 \delta\}^2 \\
+ 2(z^\top (Z^\top Z)^\dagger Z^\top X - x^\top)\beta_0 \delta^\top X_1^\top Z_1 (Z^\top Z)^\dagger z].
\end{aligned} \tag{B.26}$$

The only difference in Equation (B.4) compare to Equation (B.26) is that $\beta_0$ and $\delta$ are now fixed vectors in $\mathbb{R}^d$ instead of being random. To this end we define the space of $d \times d$ orthogonal matrices as follows:

$$\mathcal{O}(d) := \{O \in \mathbb{R}^{d \times d} : O^\top O = OO^\top = \mathbb{I}_d\}.$$

Now we consider the following change of variables for a matrix $O \in \mathcal{O}(d)$:

$$\begin{aligned}
X \mapsto XO^\top &\triangleq \bar{X}, \quad \Theta \mapsto \Theta O^\top \triangleq \bar{O} \\
\beta_0 \mapsto O\beta_0 &\triangleq \bar{\beta}_0, \quad \delta \mapsto O\delta \triangleq \bar{\delta}, \quad x \mapsto Ox \triangleq \bar{x}.
\end{aligned} \tag{B.27}$$

The key thing here is to note the following

$$\bar{Z} \triangleq \sigma(\bar{X}\bar{\Theta}^\top / \sqrt{d}) / \sqrt{d} = \sigma(X\Theta^\top / \sqrt{d}) / \sqrt{d} = Z.$$

Thus, the expression of $\hat{a}(\lambda)$ in Equation (B.3) remains unchanged. Finally, due to distributional assumption on $x$ and Lemma C.1, we have $\bar{x} \sim \mathrm{Unif}\{\mathbb{S}^{d-1}(\sqrt{d})\}$ for every $O \in \mathcal{O}(d)$. Finally, note that $\bar{x}^\top \bar{\beta}_0 = x^\top \beta$, which entails that $\mathscr{B}(\beta_0, \delta) = \bar{\mathscr{B}}(\bar{\beta}_0, \bar{\delta})$ for all $O \in \mathcal{O}(d)$, where

$$\begin{aligned}
\bar{\mathscr{B}}(\bar{\beta}_0, \bar{\delta}) = \mathbb{E}_{\bar{x} \sim P_X}[\{(\bar{z}^\top (\bar{Z}^\top \bar{Z})^\dagger \bar{Z}^\top \bar{X} - \bar{x}^\top)\bar{\beta}_0\}^2 + \{\bar{z}^\top (\bar{Z}^\top \bar{Z})^\dagger \bar{Z}_1^\top \bar{X}_1 \bar{\delta}\}^2 \\
+ 2(\bar{z}^\top (\bar{Z}^\top \bar{Z})^\dagger \bar{Z}^\top \bar{X} - \bar{x}^\top)\bar{\beta}_0 \bar{\delta}^\top \bar{X}_1^\top \bar{Z}_1 (\bar{Z}^\top \bar{Z})^\dagger \bar{z}]
\end{aligned}$$

This motivates us to consider the change of variables in Equation (B.27), when $O$ is sampled from Haar measure $\mathcal{H}_d$ on $\mathcal{O}(d)$ and independent of $X, \Theta, \epsilon$. Also, note that due to Lemma C.1 and Lemma C.2, the random vectors $\bar{\beta}_0$ and $\bar{\delta}$ satisfy the setup of Section B.1. Thus the result follows immediately by noting the fact that

$$\mathscr{B}(\beta_0, \delta) = \mathbb{E}_{O \sim \mathcal{H}_d}[\bar{\mathscr{B}}(\bar{\beta}_0, \bar{\delta})] = F_\beta^2 \mathscr{B}_\beta + F_\delta^2 \mathscr{B}_\delta,$$

where the last inequality follows from taking $\lambda \to 0$ in Equation (B.11) and Equation (B.12).

$\square$

### B.3 Limiting risk for general $\beta_0$ and $\delta$

Unlike the previous section, this section studies the general property of the asymptotic risk $R_0(\hat{a})$ when the parameters $\beta_0$ and $\delta$ are in general position. Specifically, in this section we relax the assumption that $\beta_0^\top \delta \neq 0$. Thus, the only difficulty arise in analysing the cross-covariance term $2\mathbb{E}_{x \sim P_X}[(z^\top (Z^\top Z)^\dagger Z^\top X/\sqrt{d} - x^\top)\beta_0 \delta^\top X_1^\top Z_1 (Z^\top Z)^\dagger z/\sqrt{d}]$ in Equation (3.4) as it does not vanish under the absence of orthogonality assumption. Thus, essentially it boils down to showing the results in Lemma 3.3 and Lemma 3.8.

#### B.3.1 Proof of Lemma 3.3

We begin with proof of Lemma 3.3. We recall the right Equation (3.4), i.e,

$$\mathscr{B}(\beta_0, \delta) = \mathbb{E}_{x \sim P_X}[\{(z^\top (Z^\top Z)^\dagger Z^\top X - x^\top)\beta_0\}^2 + \{z^\top (Z^\top Z)^\dagger Z_1^\top X_1 \delta\}^2$$
$$+ 2(z^\top (Z^\top Z)^\dagger Z^\top X - x^\top)\beta_0 \delta^\top X_1^\top Z_1 (Z^\top Z)^\dagger z].$$

Apart from the cross-correlation term, The first two-terms of the equation does not depend on the interaction between $\beta_0$ and $\delta$, To be precise, the first two terms are only the functions of $\beta_0$ and $\delta$ individually and does not depend on the orthogonality of $\beta_0$ and $\delta$. Thus, an analysis similar to Section B.2 yields that

$$\mathscr{B}(\beta_0, \delta) = F_\beta^2 \mathscr{B}_\beta + F_\delta^2 \mathscr{B}_\delta + 2(z^\top (Z^\top Z)^\dagger Z^\top X - x^\top)\beta_0 \delta^\top X_1^\top Z_1 (Z^\top Z)^\dagger z].$$

Thus, it only remains to analyze the cross-covariance term

$$\tilde{\mathscr{C}}_{\beta, \delta} \triangleq 2\mathbb{E}_{x \sim P_X}[(z^\top (Z^\top Z)^\dagger Z^\top X - x^\top)\beta_0 \delta^\top X_1^\top Z_1 (Z^\top Z)^\dagger z],$$

To get the desired form as in Lemma 3.3, we again introduce that random variables $\bar{\beta}_0, \bar{\delta}, \bar{X}, \bar{X}_1, \bar{Z}, \bar{Z}_1$ and $\bar{z}$ as in Section B.2. Also, define the quantity

$$\mathscr{C}_{\bar{\beta}, \bar{\delta}} \triangleq 2\mathbb{E}_{\bar{x} \sim P_X}[(\bar{z}^\top (\bar{Z}^\top \bar{Z})^\dagger \bar{Z}^\top \bar{X} - \bar{x}^\top)\bar{\beta}_0 \bar{\delta}^\top \bar{X}_1^\top \bar{Z}_1 (\bar{Z}^\top \bar{Z})^\dagger \bar{z}].$$

Following, a similar argument as in Section B.2, we have $\tilde{\mathscr{C}}_{\beta, \delta} = \mathbb{E}_{O \sim \mathcal{H}_d}\{\mathscr{C}_{\bar{\beta}, \bar{\delta}}\}$. Next, note that,

$$\tilde{\mathscr{C}}_{\beta, \delta} = 2\mathbb{E}[\text{Tr}(\bar{X}_1^\top \bar{Z}_1 (\bar{Z}^\top \bar{Z})^\dagger \bar{z}(\bar{z}^\top (\bar{Z}^\top \bar{Z})^\dagger \bar{Z}^\top \bar{X} - \bar{x}^\top)\bar{\beta}_0 \bar{\delta}^\top)].$$

Thus condition on $\bar{x}$ and $\bar{z}$ and applying Lemma C.2, the result follows, i.e.,

$$\tilde{\mathscr{C}}_{\beta, \delta} = F_{\beta, \delta} \mathscr{C}_{\beta, \delta}. \tag{B.28}$$

#### B.3.2 Proof of 3.8

Following the same arguments of Section B.1 and Section B.2, it can be shown that the limiting values of $\mathbb{E}[\mathscr{B}_\beta]$, $\mathbb{E}[\mathscr{B}_\delta]$ and $\mathbb{E}[\mathscr{V}(\tau)]$ remains the same, i.e., the results of Lemma 3.7 and first part of Lemma 3.8 remains true. Thus, it only remains to analyze the cross-covariance term

$$\tilde{\mathscr{C}}_{\beta, \delta} \triangleq 2\mathbb{E}_{x \sim P_X}[(z^\top (Z^\top Z)^\dagger Z^\top X - x^\top)\beta_0 \delta^\top X_1^\top Z_1 (Z^\top Z)^\dagger z],$$

to fully characterize the asymptotic risk asymptotically. To begin with, the vector $\delta$ can be written as a direct sum of two components $\delta_1$ and $\delta_2$, where $\delta_2$ is the orthogonal projection of $\delta$ on span($\{\beta_0\}$). In particular we have

$$\delta = \delta_1 + \delta_2, \quad \text{and} \quad \delta_2 = (\mathbb{I}_d - \beta_0 \beta_0^\top / \|\beta_0\|^2)\delta.$$

Thus, there exists $\mu \in \mathbb{R}$ such that $\delta_1 = \mu\beta_0$. Thus, $\tilde{\mathscr{C}}_{\beta,\delta}$ can be decomposed in the following way:

$$\tilde{\mathscr{C}}_{\beta,\delta} = \underbrace{2\mu\mathbb{E}_{x \sim P_X}[\{\beta_0^\top X^\top Z(Z^\top Z)^\dagger zz^\top(Z^\top Z)^\dagger Z_1^\top X_1\beta_0]}_{\mathscr{C}_{\beta,\delta}^{(1,1)}}$$

$$- \underbrace{2\mu\mathbb{E}_{x \sim P_X}[\{\beta_0^\top X_1^\top Z_1(Z^\top Z)^\dagger zx^\top\beta_0]}_{\mathscr{C}_{\beta,\delta}^{(1,2)}}$$

$$+ \underbrace{2\mathbb{E}_{x \sim P_X}[(z^\top(Z^\top Z)^\dagger Z^\top X - x^\top)\beta_0\delta_2^\top X_1^\top Z_1(Z^\top Z)^\dagger z]}_{\mathscr{C}_{\beta,\delta}^{(2)}}.$$

As $\beta_0^\top \delta_2 = 0$, following the arguments in Section B.2 we have $\mathscr{C}_{\beta,\delta}^{(2)} = 0$. To analyze the terms $\mathscr{C}_{\beta,\delta}^{(1,1)}$ and $\mathscr{C}_{\beta,\delta}^{(1,1)}$, we will gain analyze their corresponding ridge equivalents

$$\mathscr{C}_{\beta,\delta}^{(1,1)}(\lambda) := \frac{2\mu}{d}\mathbb{E}_x\left(\beta_0^\top X^\top Z\Psi\widehat{U}\Psi Z_1^\top X_1\beta_0\right),$$

$$\mathscr{C}_{\beta,\delta}^{(1,2)}(\lambda) := \frac{2\mu}{\sqrt{d}}\mathbb{E}_x\left(\beta_0^\top X_1^\top Z_1\Psi ux^\top\beta_0\right),$$

where $\Psi$ and $\widehat{U}$ are defined in Equation (B.5) and Equation (B.7) respectively and $u = \sqrt{d}z$. We focus on these terms separately. To start with, by an application of exchangeability argument we note that

$$\mathbb{E}(\mathscr{C}_{\beta,\delta}^{(1,1)}(\lambda)) = \frac{2\mu F_\beta^2}{d^2}\mathbb{E}\left[\mathrm{Tr}\left\{X^\top Z\Psi U\Psi Z_1 X_1\right\}\right]$$

$$= 2\mu F_\beta^2[n_1 f(1,1) + n_1(n_1 - 1)f(1,2) + n_0 n_1 f(1,2)].$$

Using Equation (B.21) and Equation (B.23) it can be deduced that

$$\mathbb{E}(\mathscr{C}_{\beta,\delta}^{(1,1)}(\lambda)) = 2\mu F_\beta^2 \pi\Psi_2^\star(\xi, \psi_1, \psi_2, \lambda, \mu_\star, \mu_1) + o_d(1). \tag{B.29}$$

Let us denote by $V$ the matrix $\mathbb{E}(ux^\top)$. This shows that

$$\mathbb{E}(\mathscr{C}_{\beta,\delta}^{(1,2)}(\lambda)) = \frac{2\mu}{\sqrt{d}}\mathbb{E}\left(\beta_0^\top X_1^\top Z_1\Psi V\beta_0\right).$$

Again by exchangeability argument we get

$$\mathbb{E}(\mathscr{C}_{\beta,\delta}^{(1,2)}(\lambda)) = \left(\frac{n_1}{n}\right) \cdot \frac{2\mu}{\sqrt{d}}\mathbb{E}\left(\beta_0^\top X^\top Z\Psi V\beta_0\right).$$

Let use define $T_1 := \frac{1}{\sqrt{d}}\mathbb{E}\left(\beta_0^\top X^\top Z\Psi V\beta_0\right)$. By the arguments of Section 9.1 in Mei & Montanari (2019), we can also conclude that

$$T_1 = \frac{F_\beta^2}{2}\partial_p g(i(\psi_1\psi_2\lambda)^{1/2}; \mathbf{0}) + o_d(1),$$

the function is defined in (Mei & Montanari, 2019, Equation (8.19)). Thus we have $\mathbb{E}(\mathscr{C}_{\beta,\delta}^{(1,2)}(\lambda)) = \mu F_\beta^2 \pi\partial_p g(i(\psi_1\psi_2\lambda)^{1/2}; \mathbf{0}) + o_d(1)$. This along with Equation (B.29) yields the following:

$$\mathbb{E}(\tilde{\mathscr{C}}_{\beta,\delta}) = \lim_{\lambda \to 0}\mathbb{E}(\mathscr{C}_{\beta,\delta}^{(1,1)}(\lambda)) + \mathbb{E}(\mathscr{C}_{\beta,\delta}^{(1,2)}(\lambda)) + o_d(1)$$

$$= \lim_{\lambda \to 0}\mu\pi F_\beta^2[2\Psi_2^\star(\xi, \psi_1, \psi_2, \lambda, \mu_\star, \mu_1) - \partial_p g(i(\psi_1\psi_2\lambda)^{1/2}; \mathbf{0})] + o_d(1)$$

$$= \mu\pi F_\beta^2(\mathscr{B}^\star - 1 + \Psi_2^\star) + o_d(1).$$

Finally, by a simple algebra it follows that $\mu = \frac{F_\delta}{F_\beta} \cos \phi_{\beta,\delta}$, where

$$\phi_{\beta,\delta} = \arccos\left(\frac{\langle \beta, \delta \rangle}{F_\beta F_\delta}\right).$$

This shows that $\mu F_\beta^2 = F_\beta F_\delta \cos(\phi_{\beta,\delta}) = \beta_0^\top \delta = F_{\beta,\delta}$. Now recalling Equation (B.28) we have $\lim_{d\to\infty} \mathbb{E}[\mathscr{C}_{\beta,\delta}] = \pi(\mathscr{B}^\star - 1 + \Psi_2^\star)$. This finished the proof of Lemma 3.8.

### B.4 Final form of limiting risk

Finally, Gathering all the results from Section B.1, B.2 and B.3, we have

$$\lim_{d\to\infty} \mathbb{E}(R_0(\hat{a})) = F_\beta^2 \mathscr{B}^\star + F_\delta^2 \mathscr{M}_1^\star + F_{\beta,\delta} \mathscr{M}_2^\star + \tau^2 \mathscr{V}^\star.$$

with $\mathscr{C}^\star = \pi(\mathscr{B}^\star - 1 + \Psi_2^\star)$. This concludes the proof of Theorem 3.9.

## C Uniform distribution on sphere and Haar measure

In this section we will discuss some useful results related to Uniform distribution on sphere and the Haar measure $\mathcal{H}_d$ on $\mathcal{O}(d)$.

**Lemma C.1.** *Let $U \sim Unif\{\mathbb{S}^{d-1}(\sqrt{d})\}$ and $O \in \mathcal{O}(d)$. Then $OU \sim Unif\{\mathbb{S}^{d-1}(\sqrt{d})\}$. Also, if $O \sim \mathcal{H}_d$ and is independent of $U$, then also $OU \sim Unif\{\mathbb{S}^{d-1}(\sqrt{d})\}$.*

*Proof.* Note that $U \stackrel{d}{=} \sqrt{d}G/\|G\|_2$, where $G \sim \mathbb{N}(0, \mathbb{I}_d)$. Next define $\tilde{G} := OG$. By property of Gaussian random vector we have $\tilde{G} \stackrel{d}{=} G$. Now the result follows from the following:

$$OU \stackrel{d}{=} \sqrt{d}\frac{OG}{\|G\|_2} = \sqrt{d}\frac{\tilde{G}}{\|\tilde{G}\|_2} \stackrel{d}{=} \sqrt{d}\frac{G}{\|G\|_2} \stackrel{d}{=} U.$$

Next, let $O \sim \mathcal{H}_d$ be an independent random matrix from $U$. Thus conditional on $O$, we have

$$OU \mid O \sim \text{Unif}\{\mathbb{S}^{d-1}(\sqrt{d})\}.$$

Thus, unconditionally we have $OU \sim \text{Unif}\{\mathbb{S}^{d-1}(\sqrt{d})\}$. $\qquad\square$

**Lemma C.2.** *Let $\beta, \delta \in \mathbb{R}^d$ such that $\beta^\top \delta = F_{\beta,\delta}$. Also, define the random vectors $\bar{\beta} = O\beta$ and $\bar{\delta} = O\delta$, where $O \sim \mathcal{H}_d$. Then the followings are true:*

$$\mathbb{E}(\bar{\beta}\bar{\delta}^\top) = \frac{F_{\beta,\delta}}{d}\mathbb{I}_d$$

$$\mathbb{E}(\bar{\beta}\bar{\beta}^\top)/\|\beta\|^2 = \mathbb{E}(\bar{\delta}\bar{\delta}^\top)/\|\delta\|^2 = \frac{1}{d}\mathbb{I}_d.$$

*Proof.* Let $O = (o_1, o_2, \ldots, o_d)^\top$, i.e, $o_i$ is the $i$th row of $O$. Let $\mathbf{T} = \mathbb{E}(\bar{\beta}\bar{\delta}^\top) = \mathbb{E}(O\beta\delta^\top O^\top)$. Also, by property of Haar measure, we have $\Pi O \stackrel{d}{=} O$ for any permutation matrix $\Pi$. This shows that $\{o_i\}_{i=1}^d$ are exchangeable. As a consequence we have

$$\mathbb{E}(o_i o_i^\top) = \frac{\sum_{k=1}^d \mathbb{E}(o_k o_k^\top)}{d} = \frac{\mathbb{E}(O^\top O)}{d} = \frac{1}{d}\mathbb{I}_d \quad \text{for all } i \in [d].$$

Now we are equipped to compute matrix $\mathbf{T}$. Note that

$$\mathbf{T}_{ii} = \mathbb{E}(o_i^\top \beta \delta^\top o_i) = \mathbb{E}\{\delta^\top o_i o_i^\top \beta\} = \delta^\top \beta/d = F_{\beta,\delta}/d.$$

Also, for $i \neq j$, we similarly get

$$\mathbf{T}_{ij} = \mathbb{E}\{\delta^\top o_j o_i^\top \beta\}.$$

Now again by property of Haar measure we know $O \stackrel{d}{=} (\mathbb{I}_d - 2e_i e_i^\top)O$, where $e_i$ denotes the $i$th canonical basis of $\mathbb{R}^d$. This implies that $o_j o_i^\top \stackrel{d}{=} -o_j o_i^\top$. Using, this property of $\mathcal{H}_d$, we also get $\mathbb{E}(o_j o_i^\top) = 0$. Thus, we get $\mathbf{T}_{ij} = 0$ and this concludes the proof for uncorrelatedness.

Next, we define $\mathbf{V}_\beta := \mathbb{E}(\bar\beta\bar\beta^\top)$. By a similar argument, it also follows that

$$(\mathbf{V}_\beta)_{i,j} = \frac{\|\beta\|_2^2}{d}\Delta_{ij},$$

where $\Delta_{ij}$ is the Kronecker delta function. The result for $\bar\delta$ can be shown following exactly the same recipe. □

## D Random feature regression model on California housing price dataset

We fit a random feature regression model on the "California Housing Price" dataset that contains information from the 1990 California census from a given district for predicting the median house price. The dataset has 10 columns containing summary statistics of approximately 18,000 houses. For example, it contains information like median household income, median house price (target), and ocean proximity. More details about the dataset can be found in `https://www.kaggle.com/datasets/camnugent/california-housing-prices`. In this section, we only consider a randomly chosen subset of size 2000 which is further partitioned into two groups based on the ocean proximity. The data has originally 5 groups based on the ocean proximity: (1) within 1 hour from ocean, (2) Near ocean, (3) Island, (4) Near bay, and (5) Inland. In this experiment, we merge the groups 1 and 2 to form the majority group `Near Ocean`, and we set the group `Inland` as the minority group. We drop the rest of the groups to limit our study in the 2-groups setting. Furthermore, we split the data into training (80%) and test (20%) datasets. The group proportions along with the median house value are shown in Figure 6 which clearly shows that there is a significant difference in the house price across the two groups.

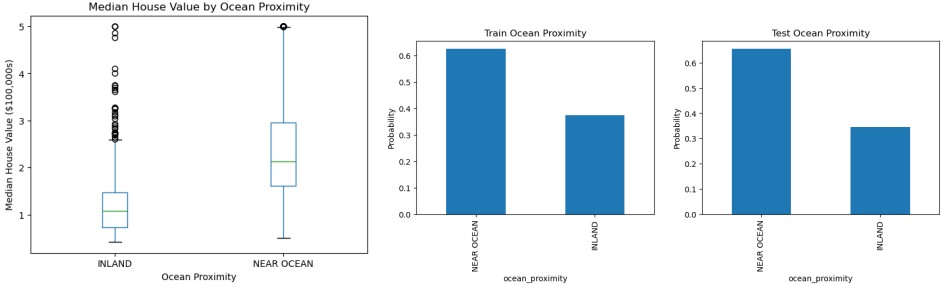

Figure 6: (left) Box plot for the median house value across two groups. (right) Histogram of ocean-proximity for training and test datasets.

We consider two cases:

1. Case 1: We coder the training dataset without any down sampling of the minority group. In this case, the majority group proportion is $\pi \approx 0.62$.

2. Case 2: We down sample the minority group in the training dataset to make the majority group proportion $\pi \approx 0.91$.

In both of the above cases, we fit the RF regression model and evaluate the mean squared error (MSE) performance on the test minority group (`Inland`) across 20 independent runs. Figure 7 shows the performance of the fitted RF model for varying choices of the overparameterization parameter $\gamma$ which clearly shows that the overparameterization ($\gamma > 1$) improves the MSE for the test minority group.

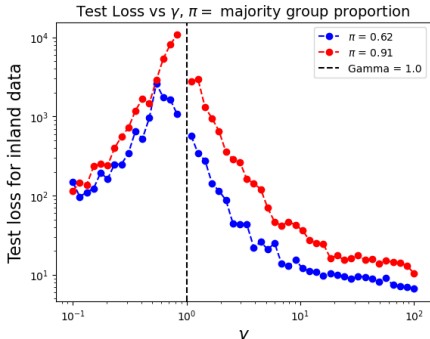

Figure 7: MSE of RF model for test minority group under varying choices of $\gamma$.

# E    Random feature classification model

In this section, we consider an overparameterized classification problem. We consider a two group setup: $g \in \{0,1\}$. Without loss of generality, we assume $\pi > \frac{1}{2}$ (so the $g = 1$ group is the majority group). The data generating process of the training examples $\{(x_i, y_i)\}_{i=1}^{n}$ is

$$
\begin{aligned}
g_i &\sim \mathsf{Ber}(\pi) \\
x_i &\sim \mathcal{N}(0, I) \\
y_i | x_i, g_i &\leftarrow \left\{ \begin{array}{ll} +1, & \text{w.p. } f(x_i^\top \beta_0)\mathbf{1}\{g_i = 0\} + f(x_i^\top \beta_1)\mathbf{1}\{g_i = 1\} \\ -1, & \text{w.p. } 1 - f(x_i^\top \beta_0)\mathbf{1}\{g_i = 0\} - f(x_i^\top \beta_1)\mathbf{1}\{g_i = 1\} \end{array} \right.
\end{aligned}
\tag{E.1}
$$

where $\beta_0, \beta_1 \in \mathbb{R}^d$ are vectors of coefficients of the minority and majority groups, respectively, and $f(t) = (1 + e^{-t})^{-1}$ is the sigmoid function.

Consider a random feature classification model, that is, for a newly generated sample $(x_{n+1}, y_{n+1})$, the classifier which predicts a label $\hat{y}_{n+1}$ for the new sample as

$$
\widehat{y} = \mathrm{sign}\left( \sum_{j=1}^{N} a_j \sigma\left( \theta_j^\top x_{n+1}/\sqrt{d} \right) \right),
$$

where $\sigma(\cdot)$ is a non-linear activation function and $N$ is the number of random features considered in the model. In the overparameterized setting ($n \le N$), we train the random feature classification model by solving the following hard-margin SVM problem:

$$
\widehat{a} \in \left\{ \begin{array}{ll} \arg\min_{a \in \mathbb{R}^N} & \|a\|_2 \\ \text{subject to} & y_i z_i^\top a \ge 1, \forall i \in [n] \end{array} \right\},
$$

where $z_i = (z_{i,1}, \dots, z_{i,N})^\top$ with $z_{i,j} = \sigma(\theta_j^\top x_i / \sqrt{d})$ for $i \in [n], j \in [N]$.

We wish to study disparity between the asymptotic risks (*i.e.*, test-time classification errors) of $\widehat{a}$ on the minority and majority groups, that is, comparing

$$
\mathcal{R}_0(\widehat{a}) = \mathbb{E}[\mathbf{1}\{\hat{y}_{n+1} \ne y_{n+1}\}|g_{n+1} = 0] \text{ and } \mathcal{R}_1(\widehat{a}) = \mathbb{E}[\mathbf{1}\{\hat{y}_{n+1} \ne y_{n+1}\}|g_{n+1} = 1]
$$

asymptotically. Moreover, we consider a high-dimensional asymptotic regime such that

$$
d \to \infty, N/d \to \psi_1 > 0, n/d \to \psi_2 > 0 \quad \text{and} \quad \gamma = \psi_1/\psi_2 = \lim_{d \to \infty} \{N/n\} \ge 1.
$$

Therefore, $\gamma$ encodes the level of overparameterization.

In the simulation for Figure 8 we let $\sigma(\cdot)$ be the ReLU activation function and $\theta_{i,j}$ be IID standard normal distributed. Moreover, we let $\pi = 0.95$, $\beta_0 = 10e_1$, $\beta_1 = 10\cos(\theta)e_1 + 10\sin(\theta)e_2$, $n = 400, d = 200, N = \gamma n$

where $e_1$ and $e_2$ are the first two standard basis of $\mathbb{R}^d$. We tune hyperparameters $\theta \in \{0°, 45°, 90°, 135°, 180°\}$ and $\gamma \in \{0.5, 1, \ldots, 3\}$, then report test errors averaged over 20 replicates.

As seen in Figure 8, the misclassification error either decreases or doesn't increase with overparameterization when the angle $\alpha \leq 90°$. This is the regime that is most related to Pham et al. (2021). On the contrary, an opposite behavior is observed for $\alpha > 90°$, when the differences across subgroups are large. The misclassification error for the minority group increases with overparameterization, which is more aligned with the observations in Sagawa et al. (2020). Interestingly, in $\alpha > 90°$ regime, undersampling improves the misclassification error for minority class compared to ERM.

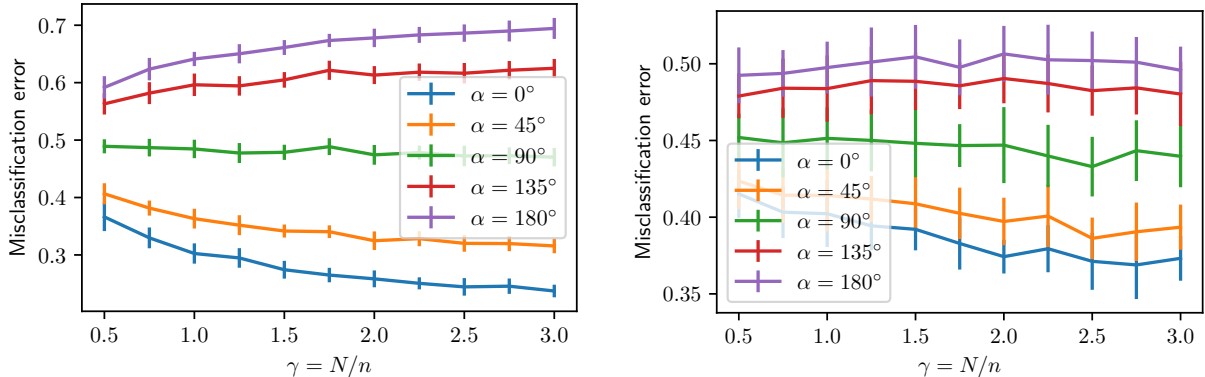

Figure 8: *Left:* misclassification errors at different levels of overparameterization under ERM. *Right:* the sample with undersampling. In each case, the errorbars are calculated over 20 repetitions.

