# OpenReview forum: "How does overparametrization affect performance on minority groups?"
_TMLR — Accepted by TMLR_

### Review · Reviewer_ZvA5 · 2025-04-13

**Summary Of Contributions:**

This paper studies the effect of overparameterization on the worst group accuracy of machine learning models. The authors analyzed this problem with a random feature regression model and derived precise asymptotic risk for the minority group. The theoretical results demonstrate a double descent curve for the minority group risk as a function of overparameterization, which suggests that increasing the model size would be beneficial to learn the minority group.

**Audience:**

Yes

**Broader Impact Concerns:**

This work raises no outstanding ethical concerns.

**Claims And Evidence:**

Yes

**Requested Changes:**

Major changes:
- The paper should better motivate the relevance of the theoretical results. For instance, do there exist empirical results for regression that highlight the poor performance of ERM where the data distribution is akin to the one considered in this paper? The literature on fairness for regression (e.g. https://proceedings.mlr.press/v97/mary19a.html, https://arxiv.org/abs/1905.12843, etc) might help to guide the process of designing a data generating process that resembles more closely the one in experimental analyses.
- Alternatively, the theoretical analysis could perhaps consider classification instead of regression, and can attempt to extend works like https://arxiv.org/abs/1803.06964 and https://arxiv.org/abs/2005.08054. In addition, the work of Sagawa et al (https://arxiv.org/abs/2005.04345) considers a simple high-dimensional setting for their theoretical results that could potentially be adapted to show how increasing overparameterization affects worst-group error for classification.
- The main takeaways of the theoretical analysis should be discussed only in the context of the problem setting. For instance, the title, abstract and introduction should reflect more clearly the key characteristics of the data distribution (i.e. regression, groups with identical support). Alternatively, more evidence should be provided to demonstrate the extent to which the assumptions hold for the experimental settings considered in Pham et al, Sagawa et al, etc.

Minor changes:
- At the end of page 1 it is stated that the works of Le Pham et al and Sagawa et al are “in sharp contrast”. My understanding  is that these papers are not contradicting each other, but rather focusing on different aspects: Sagawa et al discusses the average vs worst-group error gap (without discussing trends that emerge when varying model size), while Le Pham et al focus on how worst-group error changes with model size once we’re already in the overparameterized regime.
- in Figure 1 and Theorem 1.1, do the authors mean “Inductive bias” or “(Statistical) Bias”? If the former is true, then the paper should include a formal definition of what is understood as “Inductive bias” and how one can quantify it.

**Strengths And Weaknesses:**

Strengths
- nice (and technically sound) extension of results in Hastie et al etc to the new setting considered in the theory part of this paper. In particular, Lemma 3.8 seems to be the main theoretical contribution necessary to relate prior results to the data generating process introduced in this paper.
- Theoretical results are presented in a mathematicaly rigorous manner.

Weaknesses
- there is a stark discrepancy between the data distribution chosen to prove theoretical results in this paper and the distribution assumed in prior experimental analyses that this paper claims to attempt to explain (e.g. Pham et al; Sagawa et al; etc). In particular, two major differences stand out:
    1. this paper analyzes a regression problem, while all experimental works cited as motivation focus on classification.
    2. in the prior empirical analyses that motivate this paper (e.g. Sagawa et al), the ground truth $\beta^\star$ is indeed the minimizer of the 0-1 loss in the low noise regime, while for large noise levels a trade-off between average and worst-group performance emerges. In contrast, in this paper, neither $\beta_0$ nor $\beta_1$ are optimal for prediction, regardless of the noise regime.
    - When studying the impact of learning algorithms on worst-group error, the choice of the data distribution is crucial. For instance, depending on the distribution, improving worst-group performance may or may not come at a cost for average-case performance. Therefore, these discrepancies raise questions about how suitable the data distribution considered in this paper is for explaining prior empirical observations.

---

> ### Author Response · Authors · 2025-04-23
>
> We sincerely thank you for carefully reading the paper and providing your comments. Below, we have added responses to your concerns.
>
> ## Major comments
> **Better motivation for the theoretical results and references for poor performance of ERM:** The paper [1] shows that ERM has poor performance compared to subsampling. We'll include this reference and revise our motivation. Regarding the distributional assumption, the asymptotic regime of $d/n\to \gamma$, while yielding precise asymptotic characteristics, is quite challenging. To the best of our knowledge, we're not aware of any results that can provide such precise asymptotics under general data distributions.
>
> **Reviewer's comment regarding the extension to the classification problem:** While we agree that this direction is extremely interesting, we want to emphasize that the study of classification settings for the $d/n\to \gamma$ asymptotic regime is quite challenging and is out of the scope of this paper. Although there is some work on classification with linear models, as pointed out by the reviewer, we're not aware of any work that considers classification tasks with random features. The distinction between the linear model and the random feature model may be important, as they show different behaviors in the regression tasks. As we're not sure about the techniques required for studying classification tasks with random features, we refrain from considering classification tasks in this paper. Nonetheless, the asymptotic behaviors of our regression task align with the observed empirical behaviors in Pham et al. (2021) [2].
>
> **Addressing the comment ``The main takeaways of the theoretical analysis should be discussed only in the context of the problem setting...''** Thanks for the suggestion; we shall incorporate them in a revised version.
>
> ## Minor comments
> **Comparison between Pham et al and Sagawa et al:** We think these two papers are indeed presenting complementary results. For example, in Figure 1 of [3], the authors state that overparameterization affects minority group performance. This is not the case in [2], and the authors point this out in their discussion (2nd paragraph on Page 2).
>
> **In Figure 1 and Theorem 1.1, do the authors mean “Inductive bias” or “(Statistical) Bias”? If the former is true, then the paper should include a formal definition of what is understood as “Inductive bias” and how one can quantify it:** Thank you for the suggestion. We will clearly define the inductive bias in the revised version.
>
> ## References
>
> [1] An, J., Ying, L., & Zhu, Y. (2020). Why resampling outperforms reweighting for correcting sampling bias with stochastic gradients. arXiv preprint arXiv:2009.13447.
>
> [2] Pham, A., Chan, E., Srivatsa, V., Ghosh, D., Yang, Y., Yu, Y., ... & Steinhardt, J. (2021). The effect of model size on worst-group generalization. arXiv preprint arXiv:2112.04094.
>
> [3] Sagawa, S., Raghunathan, A., Koh, P. W., & Liang, P. (2020, November). An investigation of why overparameterization exacerbates spurious correlations. In International Conference on Machine Learning (pp. 8346-8356). PMLR.

---

> > ### Comment · Reviewer_ZvA5 · 2025-05-18
> > **Response to authors**
> >
> > I want to begin by thanking the authors for their response!
> >
> > As argued in my review, my primary concern about this paper regards the disconnect between the setting analyzed theoretically (regression instead of classification, and an inherently inconsistent problem where the ground truth is not the minimizer of the 0-1 loss) and the setting of the empirical works that the current paper claims to try to explain. In my view, the changes to the manuscript do not address these concerns. As suggested in my review, I think it is necessary that the paper either analyze a classification problem or, instead, demonstrate empirically that this phenomenon is indeed prevalent in regression with a similarly misspecified setting (i.e. with ground truth not minimizing risk).

---

### Review · Reviewer_FusR · 2025-04-15

**Summary Of Contributions:**

The paper aims to theoretically investigate the effect of overparametrization on the risk of underrepresented groups in the dataset. The paper develops its theoretical results using the ERM of linear regression and random features models. The authors then prove that:

1. Asymptotically, the expected risk of the minority group can be fully characterized when using strong assumptions on data distribution, random feature distribution, and the activation function. Using their characterization, we conclude that, in the limit, the risk in the minority group decreases with overparametrization.

2. Similarly, the paper analyzes the risk in the minority group using the linear regression ERM and concludes that overparametrization increases the risk in the minority group.

3. Finally, the authors consider using subsampling with the aim of improving the risk of minority groups. Their asymptotic characterization indicates that subsampling improves the risk for the minority group compared to just using the ERM.

**Audience:**

Yes

**Broader Impact Concerns:**

1. The conclusion in the paper seems very limited and specific to the random features model.

2. The theoretical results are interesting, but they seem like applications of the results from Montanari alongside additional algebra. It is unclear to me if there is any novel theoretical technique in the proofs that the authors would like to highlight.

3. The results hold on the limit of dimensionality, which is not the case for regressors. Therefore, it is hard to know if the results are applicable in practice.

Conclusion: the message of the paper is not surprising from a practical point of view, the theoretical results only hold in a limited scenario, and the authors didn't highlight any novel technique used in their proof.

**Claims And Evidence:**

Yes

**Requested Changes:**

1. Can the authors improve the clarity of Table 1, as discussed in the weakness?

2. Could you discuss the interpretation of variables defined in Def. 3.6.

3. Is it possible to write additional limitations discussing the assumptions necessary in your main results? Namely: Data uniformly distributed in the sphere.

4. The last paragraph in the conclusion seems speculative "using overparameterized models is unlikely to magnify disparities across groups." You only know this for RFM, and even your own results in linear regression go against this claim.

5. Is it possible to tone down the abstract as well? You argue that "we show that overparameterization always improves minority group performance." However, from my understanding, this finding is (i) limited to RFM and (ii) only asymptotic. [4 & 5 make the results not support the claims].

6. In the paragraph after Eq. 3.1, could you not use $a$ in the expectation? It is a little confusing when you have the coefficients being $a$.

7. Can you add a section describing how to compute the empirical results for Figures 2 and 3?

**Strengths And Weaknesses:**

## Strengths
1. The paper studies an interesting problem: theoretically explain the empirically observed impact of overparametrization in the minority group.

2. The paper presents interesting theoretical results.


## Weaknesses
1. The notation is not easy to follow. In fact, Table 1 is tricky to parse. For example, some of the variables defined are fixed, while others are the limit of the expression in the description. However, the text makes it easier to understand this distinction. It is also hard to parse the notation after Definition 3.4. It could be easier to understand if the authors provide some interpretation of the variables they define in Def. 3.4.

2. On the assumptions: I know that the theoretical results from Montanari that the paper relied on, for example, a very restrictive distribution on $x_i's$ -- uniform in the sphere. However, in reality, is this assumption reasonable? How much would your results change if you don't have uniformity in the sphere? Can you discuss more of the limitations of this (and the rest of the) assumption?

3. Intuitively, it makes sense that the assumption that minority and majority feature distributions are equal implies an easier-to-learn distribution for the minority. However, there might be some cases where this assumption does not hold. For example, imagine that the minority is distributed as $x_0$ with probability one and zero probability mass on the rest. The model can memorize the output to $x_0$. Therefore, the mode generalizes because the input always has only one fixed point $x_0$ and its respective memorized output. I make this point to note that maybe equal distribution is not the worst case. Hence, we need some math to prove the argument you make in Section 4.

4. All theoretical results are asymptotic. The paper argues that overparametrization helps the minority group in RFM. However, this is only shown in the limit. Should we expect this result to hold in $d <<  \infty $? If yes, why?

## Questions:
1. What do you know what to subsample? We only have access to the data $(x, y)$. You are assuming that the features are indistinguishable across the groups; therefore, we can't tell the group of $x$ by looking at it. Also, depending on the values of $\beta_0$ and $\beta_1$ it might be challenging to decide what is the minority group. How can we perform oversampling in this case?

2. In fairness, we are usually interested in *decisions* of consequences. However, the paper focuses on regression. How could you generalize your results for classification?

---

> ### Author Response · Authors · 2025-04-23
>
> ### Major comments
>
> **Interpretation of the variables they define in Eq. 3.4:** The quantities in Eq. (3.4) essentially capture the error incurred in estimating covariate associated with $\beta_0$ and $\delta$. For example, $B_\beta$ is essentially the mean squared distance (approximation error) between the true covariate $x$ (associated with $\beta_0$) and the estimated covariate $X^\top Z (Z^\top Z)^\dagger z$. Similarly, $B_\delta$ essentially captures the approximation error for the true covariate associated with $\delta$, which is the vector 0 for the minority group. The quantity $C_{\beta, \delta}$ tries to capture the similarity between the aforementioned feature difference vectors corresponding to $\beta_0$ and $\delta$ respectively.
>
> **Comments on Assumption 3.4 and limitations:** Thank you for your constructive comment regarding Assumption 3.4. We do agree that the assumption on the feature distribution may not cover the general scenarios. However, when $x$ follows an isotropic standard Gaussian, one can approximate the distribution by $\sqrt{d} {\sf Unif}(\mathbb{S}^{d-1})$. Also, in practice, one can scale the features so that they reside on some scaled version of the unit sphere. The main advantage of the assumption on the feature is that we can use certain asymptotic convergence results of random matrices which may not hold for general distributions. This is the main limitation of the current theory and is an interesting direction for future research.
>
> **Addressing the reviewer's comment involving the example on memorization:** We believe you're referring to the comment that "feature distribution between the two groups is same" in the "Summary and discussion" section.  In your example, having identical feature distribution would mean that the feature distribution of the majority group is also a trivial distribution with all mass at zero. In that case, the learned predictor would lean more toward the output for the majority group at $x_0$, i.e., the algorithm will be aiming to minimize the risk for the majority group and perform poorly on the minority group.
>
>
>
> **Validity of asymptotic result when  $d<<∞$:** Thank you for raising an important question. However, the simulation presented in the paper was done for $d = 200$, and we see that in Figure 2 the theoretical curves and the empirical curves are very close to each other. It suggests that the asymptotic behaviour kicks in even for a moderate size of $d$. However, evaluating the actual rate of convergence under such models is an interesting future research topic.
>
> **Concerns related to practicality of sub-sampling algorithm:** We agree with the reviewer that subsampling is not a practical algorithm if the group indices are not available at training time. But, often within group-robust optimization and group fairness literature such group indicators are available, where subsampling is often a popular choice of algorithm.  The purpose of including subsampling is to compare its performance with ERM, as presented in Figure 3.
>
>
> ## Minor comments
> Thank you for all the helpful comments. We will address all of the issues in the revised version. Below, we have added the response to some of the important concerns that you raised.
>
> **Interpretation of variables defined in Def. 3.6:** We do agree that the quantities defined in Def. 3.6 are very hard to parse. To the best of our knowledge, these quantities individually do not necessarily bear any interpretation and should be seen as purely analytical terms. However, these quantities help us to express  the bias and the variance terms in a succinct way as shown in Lemma 3.7, and their behaviors are plotted in Figure 2.
>
> **Is it possible to write additional limitations discussing the assumptions necessary in your main results? Namely: Data uniformly distributed in the sphere:**: Thanks for the suggestions. We will broaden our assumption implications by discussing the restrictions within the assumption in the revised version.
>
> **The last paragraph in the conclusion seems speculative "using overparameterized models is unlikely to magnify disparities across groups...":** We appreciate your comment. We intended to say that overparameterization may help us to improve the minority group performance in practice under non-linear models. However, it might not be the case always, and we also point this out in the last paragraph - "However, we warn the practitioners that overparameterization should not be confused with a method for improving the minority group performance."

---

> > ### Comment · Reviewer_FusR · 2025-05-02
> >
> > Thank you for the response.
> >
> > After reading the author's response to my and other reviewers' comments, I believe the paper could provide a foundation for understanding fairness in overparametrized RFM and linear regression, guiding future contributions to other architectures.
> >
> > Hence, after the authors address our comments (all reviewers), the paper provides enough evidence to support their claims. The changes that made me change this were:
> >
> > 1. Make the claims less speculative for  models different than RFM and LR,
> >
> > 2. Highlight the limitations of their algorithm and the data distribution assumption,
> >
> > 3. Talk about the limitations of their asymptotic finding and the challenges for the theoretical rate of convergence,
> >
> > 4. Improving their manuscript's exposition and notation (especially the points highlighted by 9wer and me),
> >
> > 5. Emphasize the limitations of their work when applied to classification problems and other observed experiments (as mentioned by ZvA5 and me).

---

### Review · Reviewer_9wer · 2025-04-16

**Summary Of Contributions:**

In this work, the authors are interested in how the use of overparameterized models impacts performance (assuming an ERM learning algorithm) on "minority groups" in the context of regression problems.

Here the notion of "group" is formulated in terms of a difference in conditional mean of the response. That is, for each sampled point, with probability $0 < \\pi < 1/2$, the mean of $y \\,\\vert\\, x$ is $\\beta\_{0}^{\\top}x$ (the "minority" case); otherwise it has mean $\\beta\_{1}^{\\top}x$ (the "majority" case). Everything else (variance, marginal distribution of $x$ etc.) is identical.

As mentioned above, the focus of this paper is on ERM learning algorithms, with a potentially overparameterized model, essentially a feed-forward neural network with one hidden layer, non-linear activation function, and $N$ units. The weights used in taking the linear combination of the raw inputs (denoted by vector $\\theta\_{j}$ for each $j=1,\\ldots,N$) are assumed to be random, and the ERM training just sets $N$ weights used to combine the outputs of the hidden layer (see their equation (2.4)).

Their main theoretical results look at the "mean squared prediction error for the minority samples", defined in their (3.1), which is the mean squared prediction error of the conditional mean (of $y$), conditioned on the group being the minority group. Their theoretical analysis is for the asymptotic case, where the number of features $d$, samples $n$, and units $N$ grow together without bound. The "overparameterized" setting is that in which $N/n$ converges to a number greater than $1$ in this limit. Their results are not for any arbitrary ERM solution, but rather the unique "minimum norm interpolator".

The main results are somewhat difficult to parse, but by breaking down the risk of interest into bias and variance terms, the authors look at how the impact of the ratio $N/n$ in the limit impacts these terms, and come to the conclusion that overparameterization should not be expected to harm the minority risk, all else equal.

**Audience:**

Yes

**Broader Impact Concerns:**

Not applicable.

**Claims And Evidence:**

No

**Requested Changes:**

Please see the comments raised above. I think the paper represents a genuine effort, but it still needs a fair bit of polish.

**Strengths And Weaknesses:**

The research problem of interest is stated clearly, and the broad area of machine learning under non-standard evaluation metrics (for example group-balanced risks under group imbalance, etc.) is definitely of importance in practice.

On the other hand, while at first glance the writing may seem fine, I feel that the presentation of the technical content in this paper is unsatisfactory at present, and below the standard of clarity that is expected for a TMLR submission (noting that TMLR emphasizes clarity above impact and significance). I will try my best to give some concrete examples to back up my criticism, breaking this up into two major points. First is poor technical presentation, and second is a lack of clarity regarding what the authors have done on top of previous literature.

__On technical presentation issues__

Throughout the technical exposition in this paper, there are countless points which I tripped up on; in my opinion this is sloppy writing. Here is a list of some examples, not in any particular order, which is not exhaustive, but I think sufficiently illustrative.

- Figure 1: it says "variance" is being compared; where is this?

- Eqn (2.1): $\\epsilon$ should be $\\varepsilon$.

- Section 2.2 and onward: $\\Theta$ is undefined.

- Section 2.2: $x\_{j}$ is used to represent individual features whereas $x\_{i}$ is used to represent vectors.

- For (2.6), writing $\\hat{a} \\in \\text{arg min}\\{\\ldots\\}$ is misleading, since the solution is unique, as shown by the equality in the same equation.

- The notation $\\hat{a}$ is used in several contexts so as to make the first equality in (2.7) quite unclear. It really should be that the right-most quantity shown in (2.6) is equal to $\\lim\_{\\lambda \\to 0\_{+}} \\hat{a}\_{\\lambda}$, right? On top of this, this "fact" is given as common knowledge, but really a precise reference would be nice.

- Before (2.8), they say "having an estimator $\\hat{a}$ of the parameter $a$...", but in reality $a$ is just a placeholder variable, not a pre-fixed unknown parameter to be estimated.

- Section 2.3.2, second paragraph: "due to its superior statistical efficiency"; is this a formal statement? A reference would be desirable.

- Table 1: $\\psi\_{1}$ and $\\psi\_{2}$ are just ratios, whereas in the main text they are the *limits* of the given ratios. This mismatch is troublesome.

- Page 6, last paragraph: "the variance ... captures the average variance of..." poor vocabulary.

- Page 6, last paragraph: what is $\\hat{f}$? A typo perhaps.

- Page 6, last paragraph: $z = \\sigma(\\Theta x/\\sqrt{d})$ appears, but $\\Theta$ is undefined, and implicitly assumed to be a sequence of vectors, making this difficult to interpret. I know what the authors have in mind, but this is sloppy.

- Lemma 3.1: why is $\\epsilon$ used here instead of $\\varepsilon$ as in the data generation section?

- Assumption 3.2: I have no idea what this assumption means, or why it is even called an assumption. How is this any different from the *definition* of $F\_{\\beta}$, $F\_{\\delta}$, and $F\_{\\beta,\\delta}$? What is being *assumed*?

- Assumption 3.5: Maybe some readers are quite familiar with this assumption, but it stood out to me. $\\sigma$ is assumed to be "weakly differentiable". This is a rather technical notion; is this degree of generality really needed? The bound on the (weak) derivative also is not something I'm familiar with. Some additional commentary would be natural considering the audience of TMLR.

- Lemmas 3.7 and 3.8: all of a sudden it seems that $\\beta\_{0}$ and $\\beta\_{1}$ (and thus $\\delta$) are random. Does this appear anywhere in the preceding assumptions or exposition of the data generating process? So $F\_{\\beta}$, $F\_{\\delta}$, and $F\_{\\beta,\\delta}$ are thus also random, correct? Or are these "lengths" and "angles" assumed to be constant in Assumption 3.2? None of this is clear.

- Section 3.1: $\\theta$ is used to denote the angle between $\\beta\_{1}$ and $\\beta\_{0}$; this clashes with previous usage of $\\theta$.

- I would say that the most critical assumption underlying the analysis in section 3 is the restriction to the minimum norm interpolator, but this is given as a nonchalant statement buried in section 2. This must be made explicit, I think (yes, I know it is stated in section 1).


__What is new here?__

Clearly much of the technical argument is rooted in previous work such as Montanari et al. (2020). Is their problem setting identical to this one? Is the main difference just that they did not analyze the same error terms as the authors have? I'm not complaining about a lack of novelty, it's a lack of clarity regarding what has been done (and learned) in the current manuscript versus what was already done and known in the closely related literature.

---

> ### Author Response · Authors · 2025-04-23
>
> ## Major comments
> **Concerns related to technical presentation and clarity:** Thank you for the constructive comments related to the presentation issue. In the updated manuscript, we will add more clarifications to the main contribution of the paper along with an improved description of the notations.
>
>
> **Intuition behind superior statistical efficiency of reweighting (Section 2.3.2, 2nd paragraph):**  While this is not a formal statement, the intuition can be understood in a simple mean estimation problem. Consider we have $m$ samples $X_{1, i}$ from the distribution $N(\theta_1, 1)$ and $n$ samples $X_{0, i}$ from the distribution $N(\theta_0, 1)$ and $m > n$. A single estimator that estimates  the means well for both distributions should estimate the parameter $\theta = 0.5 (\theta_0 + \theta_1)$. The reweighted estimate is related to $\frac1{2m}\sum {X_1, i} + \frac1{2n} \sum X_{0, i} $, which has the variance $\frac1{4m} + \frac1{4n}$. On the other hand, subsampling would down-sample the first sample to size $n$ at random and then take the average of the sample means, which has the variance $\frac1{2n}$. Clearly, the variance for the reweighting estimate is lower. This argument can be further generalized to linear regression in fixed-dimensional asymptotic regimes.
>
> **Assumption 3.5: importance of weak differentiability assumption:** We appreciate your comment. We wanted to cover a broader class of activation functions which are not necessarily differentiable. For example, the RELU activation function $R(x):= \max\{x, 0\}$ is not differentiable, but it is [weakly differentiable](https://en.wikipedia.org/wiki/Weak_derivative) with the weak derivative $R'(x) = {\bf I}(x \ge 0)$. However, $R(x)$ does satisfy the properties of Assumption 3.5. We will add more discussion in the updated version.
>
>
> ## Minor comments
> We thank the reviewer for pointing out the minor errors, including  typographical errors and presentation issues. We will address these issues in the revised version. Below, we also add responses to some of the more important ones among the minor concerns.
>
> **Concerns relate to the conflating notation  $\hat a$  in Eq. (2.7) and references:**
>   We apologize for the confusion. In the updated manuscript we will specifically use $a_{\sf mnis}$ to denote the minimum norm interpolation solution (see Section 2.3.1). We will also add the reference [1] in our discussion in Section 2.3.1.
>
>
>
> **Improving the presentation in Table 1:** We thank you for the helpful suggestion regarding the presentation of Table 1. In the revised version, we will update Table 1 to clearly point out the limiting and non-limiting quantities.
>
>
>
> **Comment on Assumption 3.2:** Thank you for spotting this. You are right about the fact that this should not be an assumption. Rather, it should be a definition. We will rectify this in the updated version.
>
>
>
> **Lemmas 3.7 and 3.8:$β_0$  and  $β_1$  (and thus  $δ$) are not random:** We apologize for the misleading writing. $\beta_0, \beta_1$ are not random variables. We will rectify this in the revised version.
>
> **Explicitly mentioning restriction to the minimum norm interpolator:** Thank you for your suggestion. We agree with you, and we will explicitly mention this at the beginning in Section 3 in the updated version.
>
> ## References
> [1] Hastie, T., Montanari, A., Rosset, S., & Tibshirani, R. J. (2022). Surprises in high-dimensional ridgeless least squares interpolation. Annals of Statistics, 50(2), 949.

---

### Author Response · Authors · 2025-04-23
**Global Response**

Thank you to all the reviewers for their time and helpful feedback.

A common concern amongst the reviewers is regarding the clarity and readability of the content of the paper. To improve the clarity of the manuscript, we will address all the issues raised by the reviewers.  In particular, we will improve the presentation of the paper in terms of delivering the main message and make a few changes in Table 1 for a better understanding of the notations.

---

### Author Response · Authors · 2025-08-13

We apologize for not communicating the new changes properly to the reviewer panel. We uploaded a revised manuscript on 29th July and specified the changes. However, we did not make an official comment for which we are extremely sorry. For reference, we reiterate the changes that we have incorporated in the latest submission. We have revised our draft with the following minor revisions as requested by the AE and the referees:

- We have added a new experiment in Appendix D featuring the California Housing dataset. Here, the target is to predict the median house price, and we find that overparameterization in a random feature regression model improves mean squared error for the minority group comprising inland houses. We point this out in the Introduction section (Page 2, Paragraph 2).

- We have provided a simulation study for the classification setting in Appendix E. We find that overparametrization indeed improved the misclassification error for the minority group when the difference between the majority and minority groups is relatively small. This matches the findings in [1]. However, under larger disparity between the majority and minority groups, we find that overparameterization aggravates the misclassification error, similar to the findings in [2]. We also mention this in the paragraph before Section 1.1 (Page 4).

- We have also added new references in the discussion in Section 1.1. The current references include some contemporary works that aim to induce fairness under the DRO, performative prediction, and active learning settings.

Lastly, we thank the AE and the reviewers for all their constructive comments, which significantly improved the quality of the work.

[1] Pham, A., Chan, E., Srivatsa, V., Ghosh, D., Yang, Y., Yu, Y., ... & Steinhardt, J. (2021). The effect of model size on worst-group generalization. arXiv preprint arXiv:2112.04094.

[2] Sagawa, S., Raghunathan, A., Koh, P. W., & Liang, P. (2020, November). An investigation of why overparameterization exacerbates spurious correlations. In International Conference on Machine Learning (pp. 8346-8356). PMLR.

---

> ### Comment · Action_Editor_B7S4 · 2025-08-14
> **Approved**
>
> Dear authors,
>
> Many thanks for diligently addressing these comments. The revisions are great and I am pleased to verify the camera-ready revision as is.
>
> Best,\
> Ahmad

---

### Decision · Action_Editor_B7S4 · 2025-05-21

**Recommendation:** Accept with minor revision

**Comment:**

The paper was reviewed by three expert reviewers on the topic. All three reviewers initially expressed concerns about the disconnect between the theoretical findings and the claims of the paper. Subsequently, the authors have revised the paper to better mirror the findings.  However, I think the paper still suffers from this disconnect as mentioned by Reviewer ZvA5 in their decision letter.

"my primary concern about this paper regards the disconnect between the setting analyzed theoretically (regression instead of classification, and an inherently inconsistent problem where the ground truth is not the minimizer of the 0-1 loss) and the setting of the empirical works that the current paper claims to try to explain. In my view, the changes to the manuscript do not address these concerns. As suggested in my review, I think it is necessary that the paper either analyze a classification problem or, instead, demonstrate empirically that this phenomenon is indeed prevalent in regression with a similarly misspecified setting (i.e. with ground truth not minimizing risk)."

Hence, I think the paper still needs another round of revision to reach the publishable state. In particular,
- Please empirically verify that a similar phenomenon to classification also appears in a real-world regression task. For example, you may want to look into the empirical setup of https://arxiv.org/abs/1905.12843 or https://arxiv.org/abs/2405.04034.
- Please update the literature survey. The paper's citations on fairness literature is out of date and sparse especially beyond 2021. It is important to make a thorough effort to contextualize the work correctly. For example, fairness regularization is not discussed at all. Please explore the last 4 years of literature in TMLR, ICML, etc and discuss their relevance. You can start with https://arxiv.org/abs/2207.07068 and https://openreview.net/forum?id=3HE4vPNIfX and the references therein.

The paper is accepted subject to the verification of these remaining issues by the AE. Congratulations!

**Audience:**

This paper studies the effect of overparameterization on group fairness and is well within the scope of TMLR.

**Claims And Evidence:**

This paper theoretically investigates the effect of model overparameterization on regression tasks with random feature kernels. Motivated by recent findings that show that model overparameterization improves the performance of minority class in classification tasks, this paper formalizes this notion and theoretically reasons about these benefits and also studies the effect of subsampling the majority group (another successful mitigation technique for group fairness). While the initial paper received many comments from the reviewers on the disconnect between the claims and the findings, the revised paper has largely addressed these issues albeit with some remaining concerns that I will detail in comments.

---

> ### Author Response · Authors · 2025-06-11
> **Minor changes to the paper**
>
> Thank you for your feedback. We have revised our draft with the following main changes:
>
> - We have provided a simulation study for the classification setting in Appendix D. We find that overparametrization indeed improved the misclassification error for the minority group when the difference between the majority and minority groups is relatively small. This matches the findings in [1]. However, under larger disparity between the majority and minority groups, we find that overparameterization aggravates the misclassification error, similar to the findings in [2]. We also mention this in the paragraph before Section 1.1 (Page 3).
>
> - We have also added new references in the discussion in Section 1.1. The current references include some contemporary works that aim to induce fairness under the DRO, performative prediction, and active learning settings.
>
> [1] Pham, A., Chan, E., Srivatsa, V., Ghosh, D., Yang, Y., Yu, Y., ... & Steinhardt, J. (2021). The effect of model size on worst-group generalization. arXiv preprint arXiv:2112.04094.
>
> [2] Sagawa, S., Raghunathan, A., Koh, P. W., & Liang, P. (2020, November). An investigation of why overparameterization exacerbates spurious correlations. In International Conference on Machine Learning (pp. 8346-8356). PMLR.

---

> > ### Comment · Action_Editor_B7S4 · 2025-07-02
> >
> > Thanks! Given that this paper studies regression, I think there is a need for an empirical verification in a regression setting as discussed in the letter. The simulation in Appendix D is nice, but does not address the issue that the phenomenon that the empirical evidence for boosting/hurting minority group performance in classification may be disconnected from the theoretical investigation performed in this paper on regression. Can the authors please address this discrepancy more carefully? If not, please add a disclaimer about this limitation in the introduction where contributions are discussed and also as a limitations subsection in the conclusion section.